# Inference from longitudinal laboratory tests characterizes temporal evolution of COVID-19-associated coagulopathy (CAC)

Colin Pawlowski[1], Tyler Wagner[1], Arjun Puranik[1], Karthik Murugadoss[1], Liam Loscalzo[1], AJ Venkatakrishnan[1], Rajiv K Pruthi[2], Damon E Houghton[2], John C O'Horo[2], William G Morice II[2,3], Amy W Williams[2], Gregory J Gores[2], John Halamka[2,4], Andrew D Badley[2], Elliot S Barnathan[5], Hideo Makimura[5], Najat Khan[5], Venky Soundararajan[1]*

[1]nference, inc, Cambridge, United States; [2]Mayo Clinic, Rochester, United States; [3]Mayo Clinic Laboratories, Rochester, United States; [4]Mayo Clinic Platform, Rochester, United States; [5]Janssen pharmaceutical companies of Johnson & Johnson (J&J), Spring House, United States

**Abstract** Temporal inference from laboratory testing results and triangulation with clinical outcomes extracted from unstructured electronic health record (EHR) provider notes is integral to advancing precision medicine. Here, we studied 246 SARS-CoV-2 PCR-positive (COVID$_{pos}$) patients and propensity-matched 2460 SARS-CoV-2 PCR-negative (COVID$_{neg}$) patients subjected to around 700,000 lab tests cumulatively across 194 assays. Compared to COVID$_{neg}$ patients at the time of diagnostic testing, COVID$_{pos}$ patients tended to have higher plasma fibrinogen levels and lower platelet counts. However, as the infection evolves, COVID$_{pos}$ patients distinctively show declining fibrinogen, increasing platelet counts, and lower white blood cell counts. Augmented curation of EHRs suggests that only a minority of COVID$_{pos}$ patients develop thromboembolism, and rarely, disseminated intravascular coagulopathy (DIC), with patients generally not displaying platelet reductions typical of consumptive coagulopathies. These temporal trends provide fine-grained resolution into COVID-19 associated coagulopathy (CAC) and set the stage for personalizing thromboprophylaxis.

*For correspondence:
venky@nference.net

## Introduction

There is a growing body of evidence suggesting that severe COVID-19 outcomes may be associated with dysregulated coagulation (*Tang et al., 2020*), including stroke, pulmonary embolism, myocardial infarction, and other venous or arterial thromboembolic complications (*Klok et al., 2020*). This so-called COVID-19 associated coagulopathy (CAC) shares similarities with disseminated intravascular coagulation (DIC) and thrombotic microangiopathy but also has distinctive features (*Levi et al., 2020*). Given the significance of CAC to COVID-19 mortality, there is an urgent need for fine-grained resolution into the temporal manifestation of CAC, particularly in comparison to the broad-spectrum of other, better characterized coagulopathies. While there are studies suggesting associations between COVID-19 infection and mortality with thrombocytopenia, D-dimer levels, and prolongation of prothrombin time, the signatures of CAC onset and progression as well as their connection to clinical outcomes are not well defined (*Tang et al., 2020*; *Gao et al., 2020*; *Panigada et al., 2020*). An advanced understanding of this phenotype may aid in the risk stratification of patients, thus facilitating optimal monitoring strategies during disease evolution through the paradigm of precision medicine.

To this end, we instituted a holistic data science platform across an academic medical center that enables machine intelligence to augment the curation of phenotypes and outcomes from over 10 million electronic health record (EHR) clinical notes and associated 3.2 million lab tests from 2232 SARS-CoV-2 positive (COVID$_{pos}$) and 72,354 confirmed SARS-CoV-2 negative (COVID$_{neg}$) patients over a retrospectively defined 2-month observation period straddling the date of the PCR test. For the COVID$_{pos}$ cohort, we center the 2-month observation period around the date of the first positive PCR test for SARS-CoV-2, and for the COVID$_{neg}$ cohort, we center the 2-month observation period around the date of the first PCR test for SARS-CoV-2 (see Materials and methods). It is important to note that not all individuals infected by SARS-CoV-2 develop symptoms of COVID-19, but rather that a majority of patients are either asymptomatic or have mild-to-moderate symptoms not requiring hospitalization for COVID-19 (*Wagner et al., 2020*). Furthermore, the guidelines followed for PCR-testing included a routine screening of individuals, patients displaying COVID-19 symptoms as per the Mayo Clinic (*Coronavirus disease, 2019*) and CDC definitions (*Website, 2020*), and possibly contact with infected persons or underlying predisposing conditions (*Wagner et al., 2020*).

By compiling all available laboratory testing data for the 30 days preceding the first SARS-CoV-2 PCR positive diagnostic testing date (day 0), as well as the 30 days following the diagnostic testing date, and triangulating this information with medications and clinical outcomes, we were able to identify laboratory abnormalities significantly associated with the COVID$_{pos}$ group. We identified coagulation-related parameters among this set of abnormalities and then studied aggregate as well as individual patient trajectories that could aid in extracting a temporal signature of CAC onset and progression. We also correlated these signals with the clinical outcomes of these patients.

In order to hone into longitudinal lab test trends that would apply at the individual patient level, we restricted our analysis to patients with available serial testing data, which had at least three test results of the same type during the observation period. After applying these inclusion criteria, 246 COVID$_{pos}$ and 13,666 COVID$_{neg}$ patients met study the inclusion criteria. The need for longitudinal data on the testing results, while constraining the study population size greatly, enables us to provide a fine-grained temporal resolution of CAC for the first time.

After filtering the patients with the available longitudinal testing data, the median age in the COVID$_{pos}$ and COVID$_{neg}$ groups were 60.8 years and 64.1 years, respectively (see Materials and methods and *Table 1*), and the numbers of males were 137 (56%) and 7129 (52%), respectively. The total numbers of pre-existing coagulopathies in the COVID$_{pos}$ and COVID$_{neg}$ groups were 31 (13%) and 3901 (29%), respectively. These counts of coagulopathies include the following phenotypes identified in the clinical notes from day −365 to day −31 relative to the PCR testing date: deep vein thrombosis, pulmonary embolism, myocardial infarction, venous thromboembolism, thrombotic stroke, cerebral venous thrombosis, and disseminated intravascular coagulation (see *Table 1* for detailed breakdown). The number of COVID$_{pos}$ patients hospitalized in the month prior to the SARS-CoV-2 PCR testing date was 41 (17%), compared to 1247 (9.1%) for the COVID$_{neg}$ cohort.

To balance these clinical covariates and others between the two cohorts, we applied 1:10 propensity score matching to define a subset of 2460 patients from the COVID$_{neg}$ cohort to use for the final statistical analysis (see Materials and methods). In particular, the general categories of covariates considered for balancing included: demographics, anticoagulant/antiplatelet medication use, medical history of pre-existing coagulopathies, and hospital admission status. Population-level characteristics of the COVID$_{pos}$, COVID$_{neg}$, and the final propensity score-matched COVID$_{neg}$ (matched) cohorts are summarized in *Table 1*. We observe that the COVID$_{pos}$ and COVID$_{neg}$ (matched) cohorts are well-balanced along these covariates which are potential confounding variables for thrombotic events and coagulopathy-related lab tests during the study period.

## Results

### Longitudinal analysis identifies lab test results characteristic of COVID-19 at specific prognostic time intervals

To identify laboratory test results that differ between COVID$_{pos}$ and COVID$_{neg}$ (matched) patients, we analyzed longitudinal trends of 194 laboratory test results in the 30 days before and after the day of PCR testing (designated as day 0). As most patients did not undergo laboratory testing for each

**Table 1.** Summary of patient characteristics for the overall COVID$_{pos}$, COVID$_{neg}$ (matched), and COVID$_{neg}$ cohorts.

The COVID$_{neg}$ (matched) cohort was constructed using 1:10 propensity score matching to balance each of the clinical covariates, including demographics (age, gender, race), medication use (anticoagulant/antiplatelet use in the preceding 30 days/1 year of PCR testing date), medical history of thrombotic events from the past year, and hospitalization status in the month prior to the date of PCR testing.

| Patient characteristics | COVID$_{pos}$ | COVID$_{neg}$ (matched) | COVID$_{neg}$ |
|---|---|---|---|
| Number of patients | 246 | 2460 | 13,666 |
| Age in years | 60.8 | 60.9 | 64.1 |
| **Gender:** | | | |
| Male | 137 (56%) | 1388 (56%) | 7129 (52%) |
| **Race:** | | | |
| White | 154 (63%) | 1540 (63%) | 12,241 (90%) |
| Black | 24 (9.8%) | 313 (13%) | 569 (4.2%) |
| Asian | 18 (7.3%) | 207 (8.4%) | 274 (2.0%) |
| American Indian | 23 (9.3%) | 81 (3.3%) | 81 (0.59%) |
| Other | 27 (11%) | 319 (13%) | 501 (3.7%) |
| **Medication use in the preceding 30 days of PCR testing date:** | | | |
| Anticoagulants | 63 (26%) | 596 (24%) | 5171 (38%) |
| Antiplatelets | 30 (12%) | 298 (12%) | 2230 (16%) |
| **Medication use in the preceding 1 year of PCR testing date:** | | | |
| Anticoagulants | 86 (35%) | 819 (33%) | 7476 (55%) |
| Antiplatelets | 40 (16%) | 419 (17%) | 3620 (26%) |
| **Medical history of thrombotic events in 1 year prior to study period:** | | | |
| Deep vein thrombosis | 15 (6.1%) | 153 (6.2%) | 2,110 (15%) |
| Pulmonary embolism | 12 (4.9%) | 112 (4.6%) | 1258 (9.2%) |
| Myocardial infarction | 11 (4.5%) | 142 (5.8%) | 1468 (11%) |
| Venous thromboembolism | 4 (1.6%) | 44 (1.8%) | 615 (4.5%) |
| Thrombotic stroke | 1 (0.41%) | 3 (0.12%) | 143 (1.0%) |
| Cerebral venous thrombosis | 0 | 1 (0.04%) | 7 (0.05%) |
| Disseminated intravascular coagulation | 0 | 1 (0.04%) | 30 (0.22%) |
| Any thrombotic event | 31 (13%) | 308 (13%) | 3901 (29%) |
| Hospitalized in the month prior to PCR testing date | 41 (17%) | 304 (12%) | 1247 (9%) |

assay on a daily basis, we grouped the measurements into nine time windows reflecting potential stages of infection as follows: pre-infection (days −30 to −11), pre-PCR (days −10 to −2), time of clinical presentation (days −1 to 0), and post-PCR phases 1 (days 1 to 3), 2 (days 4 to 6), 3 (days 7 to 9), 4 (days 10 to 12), 5 (days 13 to 15), and 6 (days 16 to 30). We only considered test-time window pairs in which at least three patients contributing to laboratory test results in both groups. During each time window, we then compared the distribution of results from COVID$_{pos}$ versus COVID$_{neg}$ (matched) patients, allowing us to identify any lab tests which were significantly altered in COVID$_{pos}$ patients during any time of disease acquisition, onset, and/or progression.

Of the 1709 lab test-time window pairs with adequate data points for comparison, we identified 130 such pairs (comprising 66 unique lab tests) which met our thresholds for statistical significance (Cohen's D >0.35, BH-adjusted Mann-Whitney p-value <0.05; *Table 2*). Among these were lab tests that may be considered positive controls for our analysis. From the time of clinical presentation onward, elevated titers of SARS-CoV-2 IgG antibodies (*Figure 1A*) and a reduction in blood oxygenation in COVID$_{pos}$ patients were observed (*Figure 1B*). We also identified abnormalities in several other classes of lab tests, including immune cell counts (*Figures 1C–E* and *2A–B*), red blood cell

**Table 2.** Summary of lab tests significantly different between COVIDpos and propensity score-matched COVIDneg cohorts during at least one clinical time window.

Data from individual patients were averaged over the defined time windows, and the mean values were compared between COVIDpos and COVIDneg patients. The lab test-time window pairs shown are those which met our defined thresholds for statistical significance and substantial effect (BH-adjusted Mann-Whitney p-value <0.05 and Cohen's D absolute value >0.35). In particular, 130 of the initial 1709 (test, time window) pairs with at least one patient met these thresholds. Rows are sorted alphabetically by test and then time window (from earliest to latest). Coagulation-related tests of particular interest (fibrinogen, platelets, prothrombin time, activated partial thromboplastin time, and D-dimer) are highlighted in gray. Sample sources are denoted as: P = plasma, S = serum, S/P = serum/plasma, B = blood, U = urine.

| Test | Units | Time window | Count COVIDpos | Count COVIDneg | Mean COVIDpos | Mean COVIDneg | Cohen's D | BH-adj M-W p-value |
|---|---|---|---|---|---|---|---|---|
| ABGRS pH Arterial | pH | Days 16–30 Post-Dx | 18 | 91 | 7.45 | 7.4 | 0.775 | 0.02 |
| ABGRS PO$_2$ Arterial | mm Hg | Days 1–3 Post-Dx | 16 | 204 | 81.9 | 129.6 | −0.797 | 3.1E-03 |
| ABGRS PO$_2$ Arterial | mm Hg | Days 4–6 Post-Dx | 25 | 82 | 78.1 | 113.2 | −0.712 | 8.8E-03 |
| ABGRS PO$_2$ Arterial | mm Hg | Days 7–9 Post-Dx | 23 | 58 | 77.2 | 121.9 | −0.807 | 1.0E-03 |
| ABGRS PO$_2$ Arterial | mm Hg | Days 10–12 Post-Dx | 18 | 37 | 76.4 | 104.2 | −0.965 | 2.6E-03 |
| ABGRS PO$_2$ Arterial | mm Hg | Days 13–15 Post-Dx | 15 | 31 | 73.1 | 112.3 | −0.964 | 6.0E-03 |
| Activated Partial Thrombopl Time, P | sec | Days 7–9 Post-Dx | 22 | 66 | 50.5 | 36.7 | 0.727 | 0.026 |
| Activated Partial Thrombopl Time, P | sec | Days 10–12 Post-Dx | 14 | 54 | 63.3 | 39.2 | 1.085 | 2.4E-03 |
| Activated Partial Thrombopl Time, P | sec | Days 13–15 Post-Dx | 16 | 48 | 53.1 | 37.6 | 1.065 | 5.6E-03 |
| Activated Partial Thrombopl Time, P | sec | Days 16–30 Post-Dx | 19 | 149 | 56.2 | 37.5 | 0.884 | 0.027 |
| Alanine Aminotransferase (ALT), P | U/L | Days 10–12 Post-Dx | 27 | 104 | 77.3 | 46 | 0.512 | 0.015 |
| Albumin, P | g/dL | Days 7–9 Post-Dx | 42 | 188 | 3.06 | 3.41 | −0.54 | 5.6E-03 |
| Albumin, S/P | g/dL | Clinical presentation | 85 | 812 | 3.43 | 3.81 | −0.591 | 4.8E-06 |
| Albumin, S/P | g/dL | Days 1–3 Post-Dx | 77 | 525 | 3.26 | 3.6 | −0.541 | 3.8E-05 |
| Albumin, S/P | g/dL | Days 10–12 Post-Dx | 61 | 254 | 3.35 | 3.66 | −0.47 | 2.6E-03 |
| Alkaline Phosphatase, P | U/L | Days 4–6 Post-Dx | 42 | 139 | 88.8 | 126.7 | −0.395 | 3.7E-03 |
| Arterial O$_2$ PP Diff | None | Clinical presentation | 21 | 106 | 268.1 | 152.1 | 0.924 | 9.7E-03 |
| Arterial O$_2$ PP Diff | None | Days 1–3 Post-Dx | 22 | 112 | 225.9 | 147.4 | 0.639 | 0.017 |
| Arterial O$_2$ PP Diff | None | Days 4–6 Post-Dx | 17 | 49 | 271.4 | 155 | 0.891 | 4.8E-03 |
| Aspartate Aminotransferase (AST), P | U/L | Days 10–12 Post-Dx | 27 | 107 | 67.6 | 44.7 | 0.404 | 3.6E-04 |
| Basophils Absolute | ×10(9)/L | Clinical presentation | 133 | 1400 | 0.0251 | 0.0379 | −0.412 | 5.8E-06 |
| Bicarbonate [MMOL/L] in Arterial Blood | mmol/L | Days 16–30 Post-Dx | 18 | 91 | 28.6 | 24.3 | 0.857 | 7.6E-03 |

*Table 2 continued on next page*

*Table 2 continued*

| Test | Units | Time window | Count COVID$_{pos}$ | Count COVID$_{neg}$ | Mean COVID$_{pos}$ | Mean COVID$_{neg}$ | Cohen's D | BH-adj M-W p-value |
|---|---|---|---|---|---|---|---|---|
| Bicarbonate in Arterial Blood | mmol/L | Days 1–3 Post-Dx | 26 | 193 | 23.2 | 21.4 | 0.513 | 0.027 |
| BUN, P | mg/dL | Days 16–30 Post-Dx | 49 | 562 | 31.4 | 21.9 | 0.555 | 3.9E-03 |
| C-reactive Protein Quantative, S | mg/L | Clinical presentation | 85 | 666 | 100.2 | 68.2 | 0.375 | 6.8E-05 |
| Calcium, Ionized, B | mg/dL | Clinical presentation | 14 | 201 | 4.36 | 4.77 | −0.67 | 0.015 |
| Calcium, Ionized, B | mg/dL | Days 1–3 Post-Dx | 18 | 270 | 4.42 | 4.73 | −0.783 | 8.5E-04 |
| Calcium, Total, P | mg/dL | Clinical presentation | 89 | 1144 | 8.71 | 9.05 | −0.468 | 5.5E-06 |
| Calcium, Total, P | mg/dL | Days 1–3 Post-Dx | 77 | 910 | 8.52 | 8.81 | −0.459 | 3.2E-04 |
| Calcium, Total, P | mg/dL | Days 7–9 Post-Dx | 71 | 353 | 8.61 | 8.93 | −0.457 | 1.8E-03 |
| Calcium, Total, S | mg/dL | Clinical presentation | 83 | 941 | 8.29 | 8.91 | −0.854 | 1.9E-13 |
| Calcium, Total, S | mg/dL | Days 1–3 Post-Dx | 98 | 1025 | 8.28 | 8.77 | −0.717 | 2.2E-10 |
| Calcium, Total, S | mg/dL | Days 4–6 Post-Dx | 87 | 568 | 8.4 | 8.69 | −0.435 | 2.3E-03 |
| Calcium, Total, S | mg/dL | Days 7–9 Post-Dx | 82 | 433 | 8.49 | 8.76 | −0.384 | 0.011 |
| Carboxyhemoglobin, ARTERIAL | % | Clinical presentation | 34 | 356 | 0.507 | 0.991 | −0.71 | 2.0E-04 |
| Carboxyhemoglobin, Arterial | % | Days 1–3 Post-Dx | 44 | 436 | 0.535 | 0.9 | −0.711 | 5.9E-05 |
| Carboxyhemoglobin, Arterial | % | Days 4–6 Post-Dx | 58 | 166 | 0.678 | 0.974 | −0.544 | 3.0E-03 |
| Carboxyhemoglobin, Arterial | % | Days 7–9 Post-Dx | 45 | 102 | 0.704 | 0.97 | −0.472 | 0.048 |
| Carboxyhemoglobin, Venous | % | Days 1–3 Post-Dx | 10 | 73 | 0.701 | 1.16 | −0.862 | 0.02 |
| Carboxyhemoglobin, Venous | % | Days 4–6 Post-Dx | 14 | 47 | 0.725 | 1.29 | −0.837 | 3.7E-03 |
| Chloride, P | mmol/L | Days 1–3 Post-Dx | 77 | 906 | 100.1 | 101.9 | −0.363 | 7.7E-03 |
| Eosinophils Absolute | ×10(9)/L | Pre-diagnosis | 28 | 547 | 0.0689 | 0.161 | −0.45 | 1.7E-03 |
| Esosinophils Absolute | ×10(9)/L | Days 4–6 Post-Dx | 133 | 559 | 0.0906 | 0.172 | −0.358 | 2.4E-06 |
| Fibrinogen, P | mg/dL | Clinical presentation | 51 | 233 | 528.9 | 360.7 | 0.859 | 8.9E-07 |
| Fibrinogen, P | mg/dL | Days 1–3 Post-Dx | 18 | 319 | 432.6 | 297.4 | 0.836 | 1.7E-03 |
| Fibrinogen, P | mg/dL | Days 4–6 Post-Dx | 26 | 116 | 477.8 | 333.7 | 0.744 | 0.014 |
| Glucose, Random, S | mg/dL | Days 13–15 Post-Dx | 49 | 314 | 150 | 126.5 | 0.544 | 0.013 |
| Hematocrit, B | % | Days 1–3 Post-Dx | 158 | 1582 | 36.5 | 33.8 | 0.433 | 9.6E-06 |

*Table 2 continued*

| Test | Units | Time window | Count COVID$_{pos}$ | Count COVID$_{neg}$ | Mean COVID$_{pos}$ | Mean COVID$_{neg}$ | Cohen's D | BH-adj M-W p-value |
|---|---|---|---|---|---|---|---|---|
| Hematocrit, B | % | Days 4–6 Post-Dx | 152 | 851 | 36 | 32.1 | 0.621 | 2.2E-10 |
| Hematocrit, B | % | Days 7–9 Post-Dx | 132 | 639 | 35.5 | 31.8 | 0.587 | 5.8E-08 |
| Hematocrit, B | % | Days 10–12 Post-Dx | 110 | 505 | 35.1 | 31.8 | 0.511 | 1.7E-05 |
| Hemoglobin Arterial | g/dL | Days 1–3 Post-Dx | 31 | 208 | 12.1 | 10.8 | 0.651 | 0.025 |
| Hemoglobin, B | g/dL | Days 1–3 Post-Dx | 158 | 1682 | 11.9 | 11.1 | 0.358 | 2.2E-04 |
| Hemoglobin, B | g/dL | Days 4–6 Post-Dx | 152 | 873 | 11.8 | 10.4 | 0.636 | 1.4E-10 |
| Hemoglobin, B | g/dL | Days 7–9 Post-Dx | 132 | 653 | 11.6 | 10.4 | 0.56 | 2.0E-07 |
| Hemoglobin, B | g/dL | Days 10–12 Post-Dx | 110 | 516 | 11.4 | 10.3 | 0.49 | 2.6E-05 |
| Ionized Calcium, Arterial | mg/dL | Days 16–30 Post-Dx | 8 | 36 | 4.93 | 4.48 | 1.561 | 0.022 |
| Lactate Dehydrogenase, S | U/L | Days 10–12 Post-Dx | 21 | 88 | 406.2 | 295.2 | 0.463 | 1.4E-03 |
| Lactate, P | mmol/L | Clinical presentation | 89 | 954 | 1.37 | 1.93 | −0.462 | 3.1E-06 |
| Lymphocytes Percent | % | Days 13–15 Post-Dx | 5 | 66 | 33.2 | 15 | 1.514 | 0.048 |
| Lymphs Absolute | ×10(9)/L | Days 13–15 Post-Dx | 56 | 349 | 3.12 | 1.11 | 0.44 | 0.018 |
| Magnesium, Plasma | mg/dL | Days 10–12 Post-Dx | 20 | 87 | 2.14 | 1.91 | 0.772 | 0.015 |
| Magnesium, S/P | mg/dL | Days 4–6 Post-Dx | 47 | 279 | 2.22 | 1.98 | 0.743 | 3.0E-03 |
| Magnesium, S/P | mg/dL | Days 7–9 Post-Dx | 40 | 215 | 2.31 | 1.97 | 1.06 | 4.1E-06 |
| Magnesium, S/P | mg/dL | Days 10–12 Post-Dx | 36 | 187 | 2.26 | 1.91 | 1.005 | 2.9E-06 |
| Magnesium, S/P | mg/dL | Days 13–15 Post-Dx | 35 | 179 | 2.22 | 1.89 | 0.904 | 1.8E-07 |
| Magnesium, S/P | mg/dL | Days 16–30 Post-Dx | 33 | 317 | 2.13 | 1.89 | 0.906 | 1.6E-04 |
| Manual Diff  Promyelocytes | % | Days 1–3 Post-Dx | 6 | 55 | 0.25 | 0 | 1.402 | 0.027 |
| Mean Corpuscular Volume | fL | Days 10–12 Post-Dx | 110 | 502 | 89.5 | 92 | −0.38 | 8.8E-03 |
| Methemoglobin, ABG | % | Clinical presentation | 34 | 356 | 0.335 | 0.571 | −0.629 | 6.0E-03 |
| Methemoglobin, ABG | % | Days 1–3 Post-Dx | 44 | 436 | 0.425 | 0.697 | −0.463 | 1.5E-03 |
| Monocytes Absolute | ×10(9)/L | Days 1–3 Post-Dx | 131 | 1079 | 0.447 | 0.748 | −0.502 | 2.6E-16 |
| Monocytes Absolute | ×10(9)/L | Days 4–6 Post-Dx | 135 | 584 | 0.475 | 0.715 | −0.597 | 2.2E-10 |
| N-terminal-PRO-Brain Type Natriuretic Peptide, S | pg/mL | Days 4–6 Post-Dx | 10 | 63 | 415.6 | 7609.7 | −0.525 | 2.9E-03 |

*Table 2 continued on next page*

*Table 2 continued*

| Test | Units | Time window | Count COVID$_{pos}$ | Count COVID$_{neg}$ | Mean COVID$_{pos}$ | Mean COVID$_{neg}$ | Cohen's D | BH-adj M-W p-value |
|---|---|---|---|---|---|---|---|---|
| Neutrophils, B | ×10(9)/L | Clinical presentation | 136 | 1382 | 5.31 | 7.12 | −0.396 | 6.3E-06 |
| Neutrophils, B | ×10(9)/L | Days 1–3 Post-Dx | 130 | 1141 | 4.73 | 6.32 | −0.385 | 5.8E-05 |
| NT-PRO BNP, P | pg/mL | Clinical presentation | 25 | 372 | 1372.4 | 5327.9 | −0.385 | 0.046 |
| NT-PRO BNP, P | pg/mL | Days 4–6 Post-Dx | 14 | 20 | 815.3 | 4388.8 | −0.929 | 0.02 |
| Nucleated RBC | /100 WBC | Days 13–15 Post-Dx | 23 | 189 | 1.24 | 0.447 | 0.561 | 1.7E-03 |
| O$_2$ HB | % | Days 1–3 Post-Dx | 13 | 242 | 88.6 | 95 | −1.37 | 2.2E-03 |
| O$_2$ HB | % | Days 4–6 Post-Dx | 32 | 90 | 92.1 | 93.7 | −0.356 | 0.013 |
| O$_2$ HB | % | Days 7–9 Post-Dx | 24 | 46 | 91.5 | 94.5 | −0.701 | 3.3E-04 |
| Osmolality, U | mOsm/kg | Pre-diagnosis | 4 | 80 | 231.5 | 478.8 | −1.509 | 0.044 |
| Oxygen Content, Arterial | vol % | Days 4–6 Post-Dx | 32 | 89 | 16 | 13.7 | 0.839 | 2.4E-03 |
| Oxygen Saturation (%) in Arterial Blood | % | Clinical presentation | 27 | 189 | 94.2 | 96.2 | −0.52 | 3.1E-03 |
| Oxygen Saturation (%) in Arterial Blood | % | Days 1–3 Post-Dx | 31 | 216 | 94.3 | 97.1 | −1.293 | 8.4E-09 |
| Oxygen Saturation (%) in Arterial Blood | % | Days 4–6 Post-Dx | 26 | 70 | 94.3 | 95.7 | −0.578 | 0.014 |
| Oxygen Saturation (%) in Arterial Blood | % | Days 10–12 Post-Dx | 18 | 29 | 93.4 | 96.5 | −1.254 | 3.1E-03 |
| Oxygen Saturation (%) in Arterial Blood | % | Days 13–15 Post-Dx | 17 | 28 | 94.8 | 96.4 | −0.671 | 0.043 |
| pH Blood Arterial | None | Days 1–3 Post-Dx | 26 | 193 | 7.42 | 7.39 | 0.539 | 0.035 |
| pH Blood Venous | pH | Days 1–3 Post-Dx | 10 | 82 | 7.42 | 7.36 | 0.963 | 0.031 |
| pH, POCT, B | None | Clinical presentation | 13 | 202 | 7.41 | 7.33 | 0.708 | 0.042 |
| Platelets | ×10(9)/L | Pre-diagnosis | 39 | 649 | 184.8 | 225.9 | −0.393 | 0.024 |
| PO$_2$ | mm Hg | Days 1–3 Post-Dx | 8 | 145 | 67.2 | 179.7 | −1.301 | 1.7E-03 |
| PO$_2$ | mm Hg | Days 7–9 Post-Dx | 14 | 16 | 71.1 | 121.1 | −0.949 | 0.027 |
| PO$_2$ Arterial | mm Hg | Days 1–3 Post-Dx | 26 | 193 | 100.4 | 150.9 | −0.87 | 8.2E-05 |
| PO$_2$ Arterial | mm Hg | Days 10–12 Post-Dx | 17 | 25 | 93.6 | 134 | −0.755 | 0.019 |
| Potassium, S | mmol/L | Pre-diagnosis | 10 | 398 | 3.93 | 4.35 | −0.836 | 0.049 |
| RABG Calculated O$_2$ Hemoglobin | % | Days 1–3 Post-Dx | 22 | 109 | 93.6 | 95 | −0.464 | 2.9E-03 |
| RABG Calculated O$_2$ Hemoglobin | % | Days 4–6 Post-Dx | 16 | 49 | 93.2 | 95.3 | −0.859 | 2.3E-03 |
| RABG Calculated O$_2$ Hemoglobin | % | Days 10–12 Post-Dx | 13 | 22 | 94 | 96.3 | −1.269 | 0.038 |

*Table 2 continued on next page*

*Table 2 continued*

| Test | Units | Time window | Count COVID$_{pos}$ | Count COVID$_{neg}$ | Mean COVID$_{pos}$ | Mean COVID$_{neg}$ | Cohen's D | BH-adj M-W p-value |
|---|---|---|---|---|---|---|---|---|
| RABG PF Ratio | None | Days 4–6 Post-Dx | 17 | 49 | 1.46 | 2.68 | −1.489 | 6.9E-05 |
| RABG PF Ratio | None | Days 7–9 Post-Dx | 13 | 22 | 1.75 | 2.56 | −1.006 | 0.038 |
| RABG PF Ratio | None | Days 10–12 Post-Dx | 13 | 22 | 1.83 | 3.22 | −1.518 | 3.9E-03 |
| RBC (Red Blood Cell) Count | ×10 (12)/L | Clinical presentation | 151 | 1671 | 4.32 | 3.99 | 0.409 | 2.0E-04 |
| RBC (Red Blood Cell) Count | ×10 (12)/L | Days 1–3 Post-Dx | 158 | 1562 | 4.13 | 3.73 | 0.524 | 5.8E-08 |
| RBC (Red Blood Cell) Count | ×10 (12)/L | Days 4–6 Post-Dx | 152 | 846 | 4.08 | 3.55 | 0.693 | 3.2E-12 |
| RBC (Red Blood Cell) Count | ×10 (12)/L | Days 7–9 Post-Dx | 132 | 635 | 4 | 3.49 | 0.656 | 2.4E-09 |
| RBC (Red Blood Cell) Count | ×10 (12)/L | Days 10–12 Post-Dx | 110 | 502 | 3.95 | 3.48 | 0.587 | 6.1E-07 |
| Red Cell Distribution Width CV | % | Days 4–6 Post-Dx | 137 | 722 | 14.1 | 15.1 | −0.373 | 3.4E-04 |
| Red Cell Distribution Width CV | % | Days 7–9 Post-Dx | 119 | 552 | 14.2 | 15.4 | −0.431 | 9.8E-05 |
| Red Cell Distribution Width CV | % | Days 10–12 Post-Dx | 97 | 429 | 14.5 | 15.7 | −0.394 | 1.2E-03 |
| Sodium, P | mmol/L | Clinical presentation | 89 | 1141 | 135.6 | 137.3 | −0.375 | 7.3E-03 |
| Sodium, P | mmol/L | Days 1–3 Post-Dx | 77 | 927 | 136.6 | 138.1 | −0.377 | 4.7E-03 |
| Sodium, S | mmol/L | Days 10–12 Post-Dx | 69 | 334 | 140.8 | 138.3 | 0.651 | 2.0E-04 |
| Spont. Breaths/min | None | Days 4–6 Post-Dx | 23 | 67 | 25 | 20.2 | 0.767 | 0.016 |
| Tacrolimus, B | ng/mL | Days 7–9 Post-Dx | 8 | 81 | 4.22 | 8.12 | −1.102 | 8.8E-03 |
| Tacrolimus, B | ng/mL | Days 10–12 Post-Dx | 8 | 79 | 3.8 | 9.24 | −1.468 | 2.5E-03 |
| Tacrolimus, B | ng/mL | Days 13–15 Post-Dx | 7 | 71 | 3.7 | 8.52 | −1.47 | 7.5E-03 |
| Tacrolimus, B | ng/mL | Days 16–30 Post-Dx | 10 | 110 | 4.93 | 7.8 | −1.094 | 0.022 |
| Temperature | None | Clinical presentation | 23 | 136 | 37 | 36.7 | 0.591 | 0.042 |
| Temperature | None | Days 1–3 Post-Dx | 23 | 189 | 37 | 36.4 | 0.765 | 4.8E-04 |
| Triglycerides, S/P | mg/dL | Days 4–6 Post-Dx | 16 | 41 | 326.2 | 173 | 1.196 | 7.3E-03 |
| Triglycerides, S/P | mg/dL | Days 7–9 Post-Dx | 17 | 24 | 310.6 | 191.5 | 0.945 | 0.016 |
| Triglycerides, S/P | mg/dL | Days 10–12 Post-Dx | 17 | 35 | 364.5 | 174.4 | 1.217 | 4.0E-03 |
| Triglycerides, S/P | mg/dL | Days 16–30 Post-Dx | 10 | 77 | 276.1 | 166.4 | 0.83 | 0.024 |
| Troponin T, 5TH GEN, P | ng/L | Days 4–6 Post-Dx | 18 | 54 | 21.4 | 245.3 | −0.499 | 7.5E-03 |

*Table 2 continued*

| Test | Units | Time window | Count COVID$_{pos}$ | Count COVID$_{neg}$ | Mean COVID$_{pos}$ | Mean COVID$_{neg}$ | Cohen's D | BH-adj M-W p-value |
|---|---|---|---|---|---|---|---|---|
| Troponin T, Baseline, 5TH Gen, P | ng/L | Days 7–9 Post-Dx | 11 | 43 | 15.1 | 53.7 | −0.538 | 0.037 |
| VBGRS HGB | g/dL | Days 4–6 Post-Dx | 36 | 99 | 12.3 | 10.5 | 0.932 | 3.6E-04 |
| White Blood Cells | ×10(9)/L | Days 1–3 Post-Dx | 158 | 1650 | 6.67 | 9.08 | −0.439 | 3.2E-12 |

counts (*Figure 2C*), mean corpuscular volume (*Figure 2D*), calcium and magnesium levels (*Figure 2E–F*), and coagulation-related tests (*Figure 3*).

With respect to coagulation, we found that plasma fibrinogen was significantly elevated in COVID$_{pos}$ patients at the time of diagnosis (Cohen's D = 0.859, BH-adjusted Mann-Whitney p-value = 8.9e-7, *Table 2*, *Figure 3A*). This hyperfibrinogenemia generally resolved during the 7 days following diagnosis (*Figure 3A*). Conversely, platelet counts were lower in the COVID$_{pos}$ cohort at the time of clinical presentation but tended to increase over the subsequent 10 days to levels significantly higher than those in COVID$_{neg}$ patients (Cohen's D = 0.229, BH-adjusted Mann-Whitney p-value = 3.6e-3, *Table 2*, *Figure 3B*). While thrombocytopenia has been reported in COVID-19 patients before (*Xu et al., 2020*; *Yang et al., 2020*), an upward trend in platelet counts after diagnosis has not been described to our knowledge. We observe extended prothrombin times in both the COVID$_{pos}$ and COVID$_{neg}$ (matched) cohorts significantly above the normal range; however, there was no differentiation between the cohorts. We observe extended activated partial thromboplastin times (aPTT) in the COVID$_{pos}$ significantly above normal levels from day 7 onward (*Figure 3D*). D-dimer levels were frequently above normal limits in both the COVID$_{pos}$ and COVID$_{neg}$ cohorts and were not significantly different between these cohorts during any time window (*Figure 3E*). The above trends hold up even when the time windows are perturbed (*Table 3*).

We also performed similar analyses comparing the COVID$_{pos}$ and COVID$_{neg}$ (matched) cohorts using different time window definitions including daily trends (*Figure 4*). This approach offers the advantage of increased granularity at the cost of sample size per time point, but we did identify similar lab tests as altered in COVID$_{pos}$ patients using each approach including the fibrinogen decline and platelet increase in the COVID$_{pos}$ cohort after diagnosis (*Figure 4*).

## Thrombosis is enriched among COVID-19 patients undergoing longitudinal lab testing

Given the recently described coagulopathies associated with COVID-19 (*Tang et al., 2020*; *Klok et al., 2020*; *Levi et al., 2020*), we were intrigued by the temporal trends in fibrinogen levels and platelet counts in the COVID$_{pos}$ cohort (*Figure 3*). Next, we asked whether the observed coagulation-related laboratory trends were associated with clinical manifestations of thrombosis. To do so, we employed a BERT-based neural network (*Devlin et al., 2018*; see Materials and methods) to identify patients who experienced a thrombotic event after their SARS-CoV-2 PCR testing date. Specifically, we extracted diagnostic sentiment from EHR notes (e.g. whether a patient was diagnosed with a phenotype, suspected of having a phenotype, ruled out for having a phenotype, or other) regarding specific thromboembolic phenotypes including deep vein thrombosis, pulmonary embolism, myocardial infarction, venous thromboembolism, thrombotic stroke, cerebral venous thrombosis, and disseminated intravascular coagulation.

We found that 101 of the total 2232 COVID$_{pos}$ cohort (4.5%) were positively diagnosed with one or more of the above-mentioned thrombotic phenotypes in the 30 days after PCR testing, with the majority of these patients (53 of 101) experiencing a deep vein thrombosis. Interestingly, we found that after creating subsets of the patients with longitudinal lab testing data (i.e. the patients meeting the criteria for inclusion in our study), 76 of the 246 patients (31%) had at least one EHR-derived clot diagnosis, including 47 patients with deep vein thrombosis (*Table 4*). Thus, the cohort under consideration here is highly enriched (*Table 5*; hypergeometric p-value $<1\times10^{-50}$) for patients experiencing thrombotic events compared to the overall COVID$_{pos}$ cohort.

**Table 3.** Sensitivity analysis of clinical time intervals for significant coagulation-related lab test trends.

Results from sensitivity analysis perturbing the time intervals for the significant (coagulation-related lab test, time interval) pairs (i.e. highlighted rows of *Table 2*). Perturbed results that met both of the significance thresholds (BH-adjusted Mann-Whitney p-value <0.05 and Cohen's D absolute value >0.35) are highlighted in light green, and perturbed results that only met one of the thresholds for either effect size or statistical significance are highlighted in yellow.

| Test | Units | Perturbation | Original time window | Count COVID$_{pos}$ | Count COVID$_{neg}$ | Mean COVID$_{pos}$ | Mean COVID$_{neg}$ | Cohen's D | BH-adjusted M-W p-value |
|---|---|---|---|---|---|---|---|---|---|
| Activated Partial Thrombopl Time, P | sec | −1 day | Days 7–9 Post-Dx | 26 | 72 | 50.1 | 38 | 0.57 | 0.034 |
| Activated Partial Thrombopl Time, P | sec | +1 day | Days 7–9 Post-Dx | 17 | 58 | 55 | 37.5 | 0.81 | 0.014 |
| Activated Partial Thrombopl Time, P | sec | −1 day | Days 10–12 Post-Dx | 16 | 57 | 56.9 | 38.4 | 0.808 | 9.10E-03 |
| Activated Partial Thrombopl Time, P | sec | +1 day | Days 10–12 Post-Dx | 15 | 60 | 56.9 | 38 | 1.106 | 2.60E-03 |
| Activated Partial Thrombopl Time, P | sec | −1 day | Days 13–15 Post-Dx | 15 | 52 | 55.5 | 37.8 | 1.041 | 0.014 |
| Activated Partial Thrombopl TIME, P | sec | +1 day | Days 13–15 Post-Dx | 14 | 48 | 51.8 | 37.1 | 0.962 | 0.015 |
| Activated Partial Thrombopl Time, P | sec | −1 day | Days 16–30 Post-Dx | 22 | 156 | 55.2 | 37 | 0.913 | 5.70E-03 |
| Activated Partial Thrombopl Time, P | sec | +1 day | Days 16–30 Post-Dx | 19 | 139 | 56 | 38.2 | 0.725 | 3.80E-02 |
| Fibrinogen, P | mg/dL | −1 day | Clinical presentation | 25 | 92 | 584.9 | 370.7 | 1.067 | 1.20E-04 |
| Fibrinogen, P | mg/dL | +1 day | Clinical presentation | 37 | 292 | 488.2 | 326.2 | 0.885 | 8.80E-06 |
| Fibrinogen, P | mg/dL | −1 day | Days 1–3 Post-Dx | 41 | 381 | 494.5 | 318 | 1.023 | 3.90E-07 |
| Fibrinogen, P | mg/dL | +1 day | Days 1–3 Post-Dx | 21 | 244 | 420.3 | 312.2 | 0.616 | 7.90E-03 |
| Fibrinogen, P | mg/dL | −1 day | Days 4–6 Post-Dx | 27 | 156 | 432.2 | 336 | 0.495 | 0.045 |
| Fibrinogen, P | mg/dL | +1 day | Days 4–6 Post-Dx | 24 | 105 | 472.2 | 333.2 | 0.712 | 0.025 |
| Platelets | x10(9)/L | −1 day | Pre-diagnosis | 34 | 575 | 187.3 | 225.6 | -0.357 | 0.057 |
| Platelets | x10(9)/L | +1 day | Pre-diagnosis | 118 | 1533 | 201.3 | 234.4 | -0.328 | 7.30E-04 |

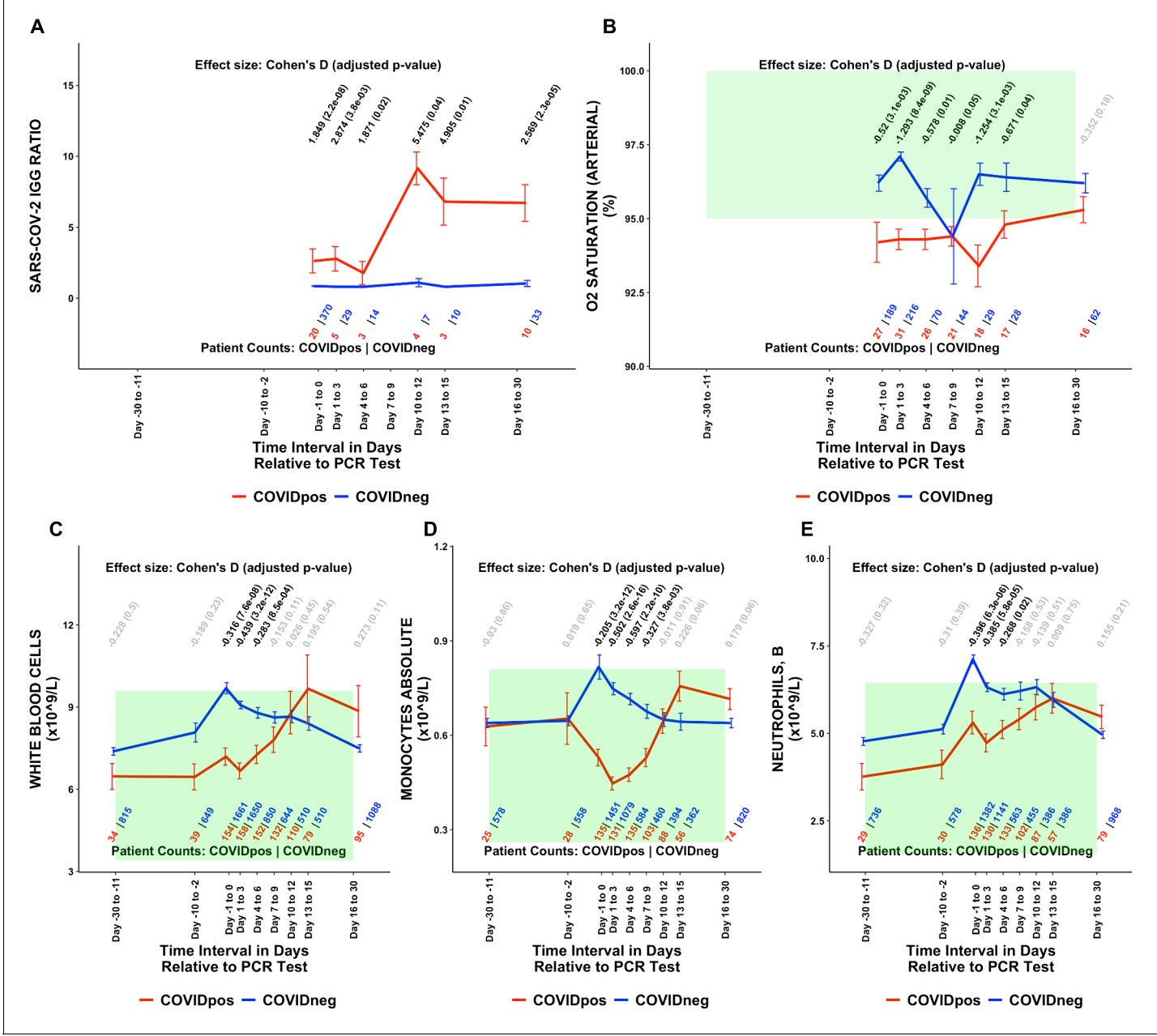

**Figure 1.** Longitudinal and temporally resolved analysis highlights positive control lab tests elevated in COVIDpos patients along with distinctive immune signatures. Longitudinal trends in COVIDpos versus COVIDneg (matched) patients for the following lab tests: (A) SARS-CoV-2 IGG ratio, (B) oxygen saturation in arterial blood, (C) white blood cells, (D) monocytes absolute, and (E) neutrophils, blood. For any window of time during which at least three patients in each cohort had test results, data are shown as mean with standard errors. The normal range for each lab test is shaded in green. Values given horizontally along the top of the plot are Cohen's D statistics comparing the COVIDpos and COVIDneg (matched) cohorts along with the BH-adjusted Mann-Whitney test p-values. Significant differences (adjusted p-value <0.05) are shown in black, while non-significant values are shown in gray. Values given horizontally along the bottom of the plot are the numbers of patients in the COVIDpos and COVIDneg cohorts, respectively (i.e. # COVIDpos | # COVIDneg). For certain lab tests, some data points are missing because these time windows had fewer than three data points in the COVIDpos cohort.

## Longitudinal platelet count trends are not strongly associated with the development of thrombosis in COVID-19 patients

Among the 246 COVIDpos patients with longitudinal lab testing data, 81 were serially tested starting at clinical presentation for fibrinogen versus 245 tested for platelets. As such, we first analyzed

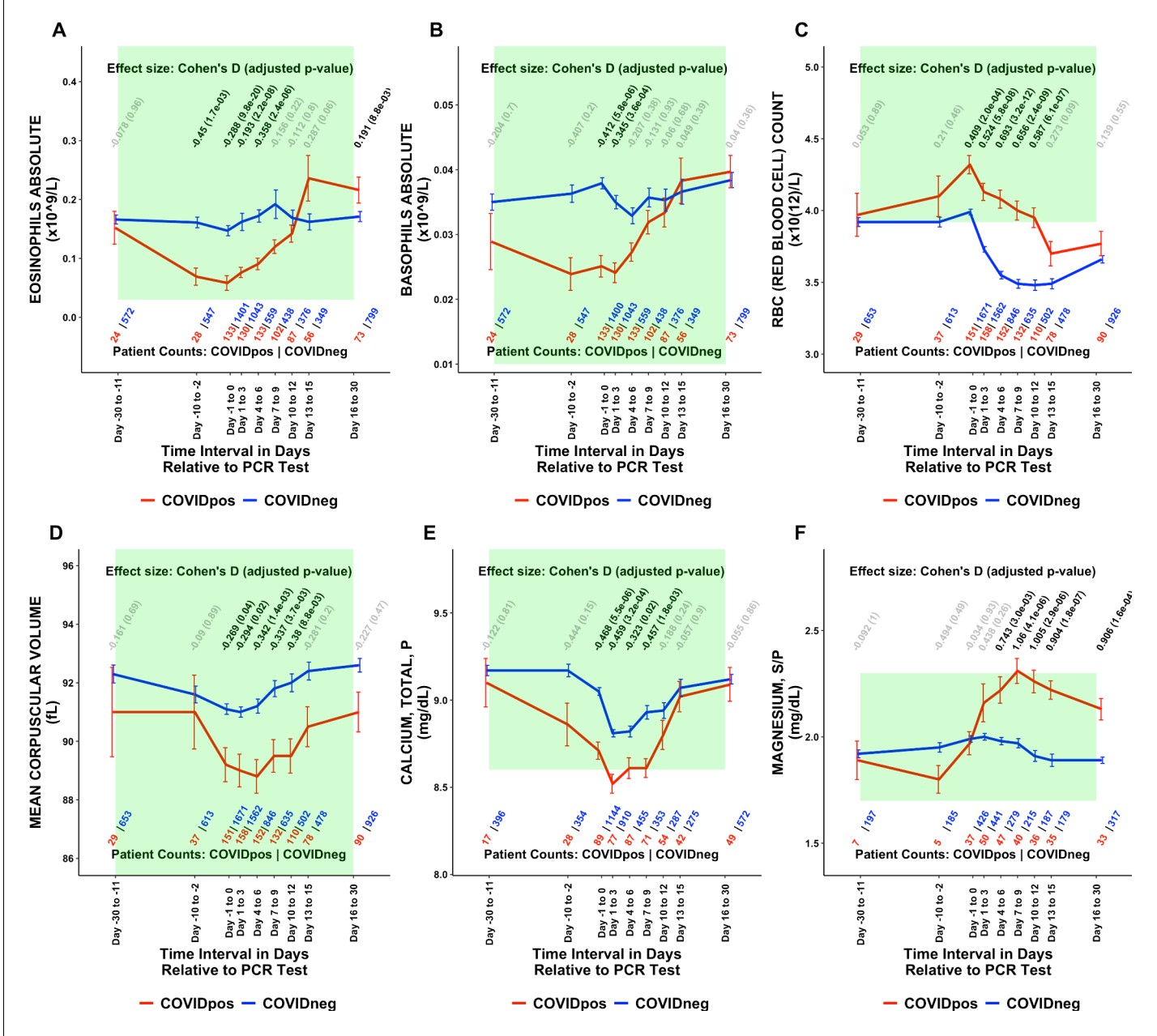

**Figure 2.** Longitudinal trends of COVID_pos patients' lab tests show distinctive immune, hematologic, and serum chemistry signatures within normal ranges. Longitudinal trends in COVID_pos versus COVID_neg (matched) patients for the following lab tests: (A) eosinophils absolute, (B) basophils absolute, (C) red blood cell count, (D) mean corpuscular volume, (E) calcium total, plasma, and (F) magnesium total, serum/plasma. For any window of time during which at least three patients in each cohort had test results, data are shown as mean with standard errors. The normal range for each lab test is shaded in green. Values given horizontally along the top of the plot are Cohen's D statistics comparing the COVID_pos and COVID_neg (matched) cohorts along with the BH-adjusted Mann-Whitney test p-values. Significant differences (adjusted p-value <0.05) are shown in black, while non-significant values are shown in gray. Values given horizontally along the bottom of the plot are the numbers of patients in the COVID_pos and COVID_neg cohorts, respectively (i.e. # COVID_pos | # COVID_neg). For certain lab tests, some data points are missing because these time windows had fewer than three data points in the COVID_pos cohort.

whether associations exist between platelet counts (or temporal alterations thereof) and clotting propensity in this cohort. Among these 245, there were 169 patients without thrombosis after PCR-based diagnosis (non-thrombotic) and 76 patients with thrombosis (thrombotic). There is a statistically significant difference between the COVID_pos and COVID_neg cohorts in the platelet count at

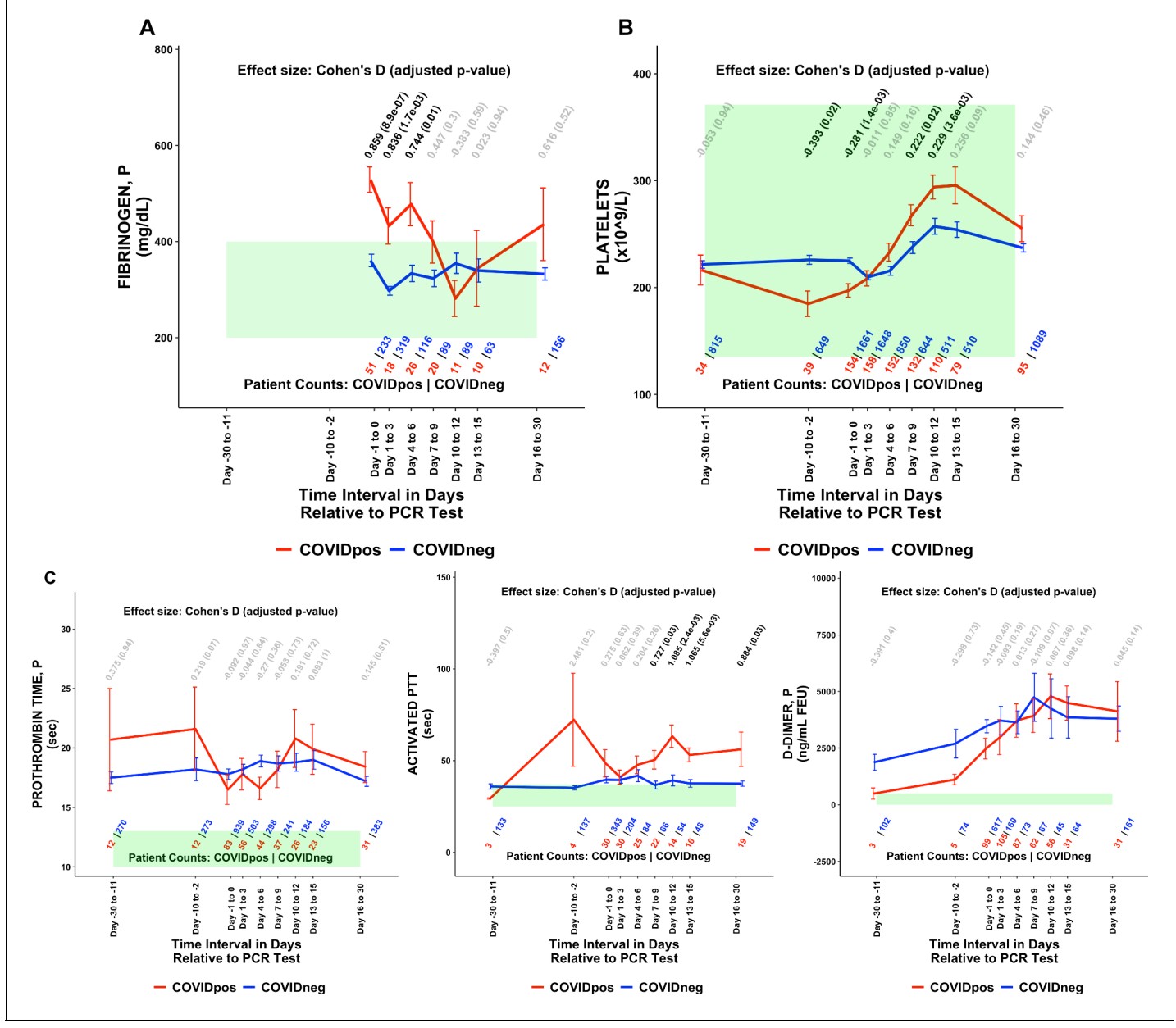

**Figure 3.** COVID$_{pos}$ patients show distinctly opposite temporal trends in fibrinogen and platelet counts starting at the time of diagnosis. Longitudinal trends of COVID$_{pos}$ versus COVID$_{neg}$ (matched) patients for the following lab tests: (A) fibrinogen, plasma, (B) platelets, and (C) other coagulation-related tests including prothrombin time (PT), activated partial thromboplastic time (aPTT), and D-dimers. For any window of time during which at least three patients in each cohort had test results, data are shown as mean with standard errors. The normal range for each lab test is shaded in green. Values given horizontally along the top of the plot are Cohen's D statistics comparing the COVID$_{pos}$ and COVID$_{neg}$ (matched) cohorts along with the BH-adjusted Mann-Whitney test p-values. Significant differences (adjusted p-value <0.05) are shown in black, while non-significant values are shown in gray. Values given horizontally along the bottom of the plot are the numbers of patients in the COVID$_{pos}$ and COVID$_{neg}$ cohorts, respectively (i.e. # COVID$_{pos}$ | # COVID$_{neg}$). For certain lab tests, some data points are missing because these time windows had fewer than three data points in the COVID$_{pos}$ cohort.

clinical presentation (*Figure 5A*). In particular, thrombocytopenia (platelet count <150×10$^9$/L) was observed in 29% (46 out of 154) COVID$_{pos}$ and 21% (346 of 1661) COVID$_{neg}$ patients at the time of diagnosis (*Figure 5A*). However, the platelet levels at this time point were not associated with the subsequent formation of a blood clot in the COVID$_{pos}$ cohort (*Figure 5B*).

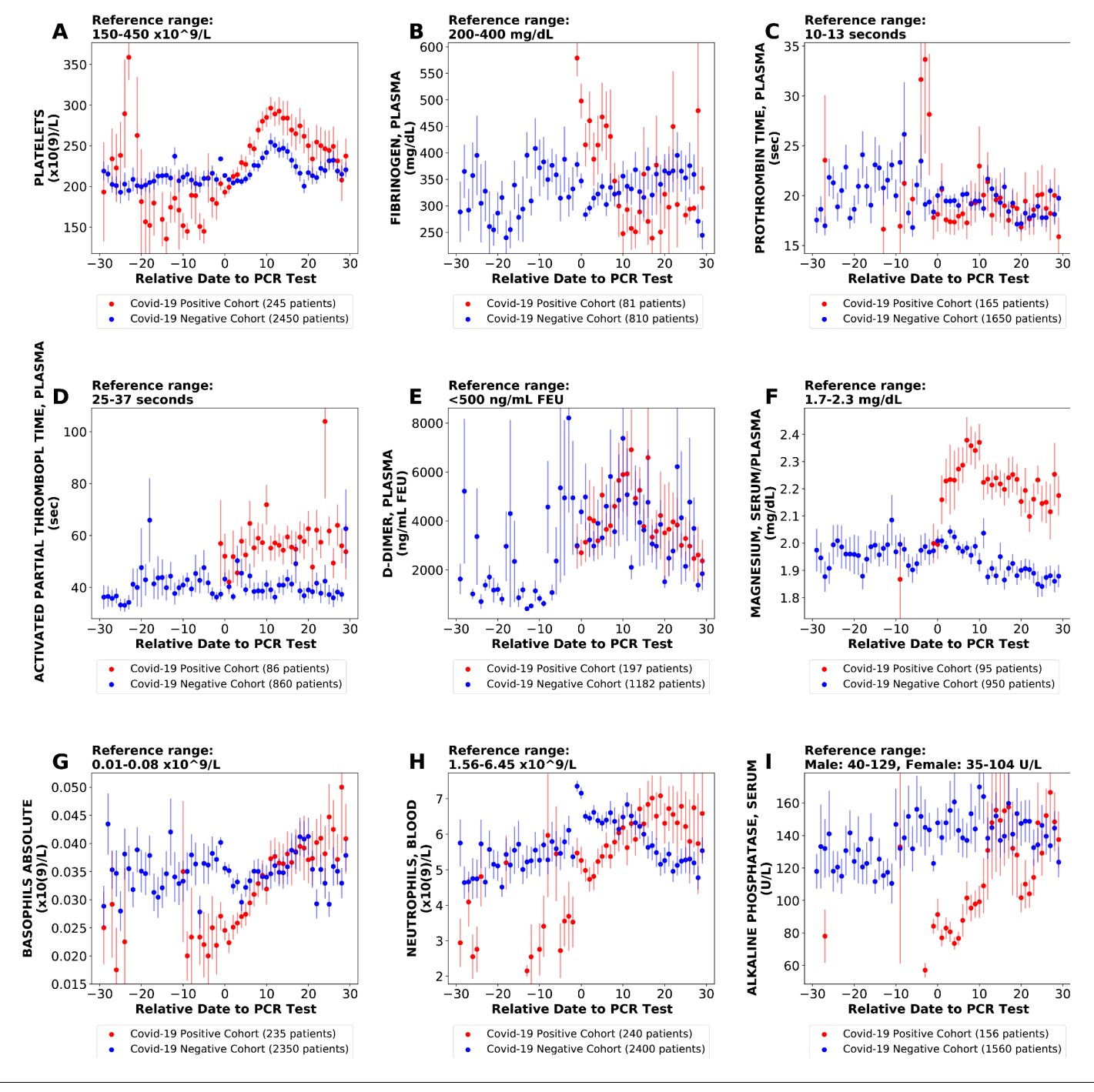

**Figure 4.** Longitudinal trends of lab tests with daily resolution. Longitudinal trends of COVID_pos versus COVID_neg (matched) patients for the following lab tests: (A) platelets; (B) fibrinogen, plasma; (C) prothrombin time, plasma; (D) activated partial thromboplastin time; (E) D-dimer; (F) magnesium, serum/plasma; (G) basophils absolute; (H) neutrophils, blood; (I) alkaline phosphatase, serum. The reference ranges are shown at the top of each plot. For each cohort, average lab values and standard errors are shown for each day with at least three observations. For certain lab tests, some data points are missing because these days had fewer than three data points in the COVID_pos cohort.

We hypothesized that the previously discussed increase in platelet counts after COVID-19 diagnosis may be associated with the development of blood clots. If true, then we would expect the thrombotic COVID_pos cohort to show significantly higher maximum platelet counts during their course of disease progression compared to the non-thrombotic COVID_pos cohort. We found that this was not

**Table 4.** Prevalence of thrombotic phenotypes after the clinical presentation in COVID$_{pos}$ patients with and without available longitudinal lab testing data.

For each clotting phenotype listed, a BERT-based neural network was used to extract diagnostic sentiment from individual EHR patient notes in which the phenotype (or a synonym thereof) was present. This automated curation was applied to clinical notes for each patient from day = −1 (clinical presentation) to day = 30 (end of the study period) relative to the PCR testing date. In this table, we show the absolute number of patients with each phenotype along with the percentage of patients in each cohort with the given specific thrombotic phenotype in parentheses.

| Clotting phenotype | Cohort 1: COVID$_{pos}$ with longitudinal data | Cohort 2: COVID$_{pos}$ without longitudinal data | Cohort 3: Complete COVID$_{pos}$ cohort |
|---|---|---|---|
| Deep vein thrombosis | 47 (19%) | 6 (0.30%) | 53 (2.4%) |
| Pulmonary embolism | 22 (8.9%) | 9 (0.45%) | 31 (1.4%) |
| Myocardial infarction | 10 (4.1%) | 8 (0.40%) | 18 (0.81%) |
| Venous thromboembolism | 7 (2.8%) | 0 | 7 (0.31%) |
| Thrombotic stroke | 2 (0.81%) | 2 (0.10%] | 4 (0.18%) |
| Cerebral venous thrombosis | 0 | 0 | 0 |
| Disseminated intravascular coagulation | 5 (2.0%) | 0 | 5 (0.22%) |
| Total unique patients with clot | 76 (31%) | 25 (1.3%) | 101 (4.5%) |
| Total patients | 246 | 1986 | 2232 |

the case, as maximum platelet counts were similar in the two groups (**Figure 5C**). Similarly, among the 147 COVID$_{pos}$ patients with platelet counts both at the time of clinical presentation and post-diagnosis, the degree of maximal platelet increase was not associated with the development of thrombosis (**Figure 5E**). It would certainly be of interest to perform this same analysis on a larger COVID$_{pos}$ cohort (n = 2232; 101 thrombotic vs. 2131 non-thrombotic), but we were not able to do so given the lack of longitudinal testing available for a large majority of non-thrombotic COVID$_{pos}$ patients (**Table 4**).

Conversely, we explored whether some COVID$_{pos}$ patients may experience clotting in the setting of low or declining platelets (e.g. consumptive coagulopathy) despite the population-level trend of increasing platelets over time. Indeed, we found that nine of 74 thrombotic patients showed absolute platelet counts below $100 \times 10^9$/L during at least one post-diagnosis time window (below dotted red line in **Figure 5D**). In addition, we analyzed post-diagnosis platelet reductions among COVID$_{pos}$ patients. While the maximum degree of absolute platelet reduction was not associated with clot development in aggregate (**Figure 5F**), we did find that six of the 52 thrombotic patients experienced a reduction of at least $100 \times 10^9$/L relative to the time of diagnosis. Of note, similar fractions of non-thrombotic COVID$_{pos}$ patients also showed these low or declining platelet counts, indicating that these trends are not specific indicators of thrombosis (**Figure 5D,F**).

**Table 5.** Enrichment of thrombotic phenotypes among COVID$_{pos}$ patients with longitudinal lab testing data.

Contingency table to calculate hypergeometric enrichment significance of thrombosis among patients with longitudinal lab testing data. The 246 patients with longitudinal testing data are those considered in this study, while the 1986 patients who did not have at least three results from one lab test over the defined 60-day window were excluded from this longitudinal analysis.

| | Patient has longitudinal data | Patient does NOT have longitudinal data | Total |
|---|---|---|---|
| Thrombosis | 76 | 25 | 101 |
| No thrombosis | 170 | 1961 | 2131 |
| Total | 246 | 1986 | 2232 |

Hypergeometric enrichment: p-value $<1 \times 10^{-50}$.

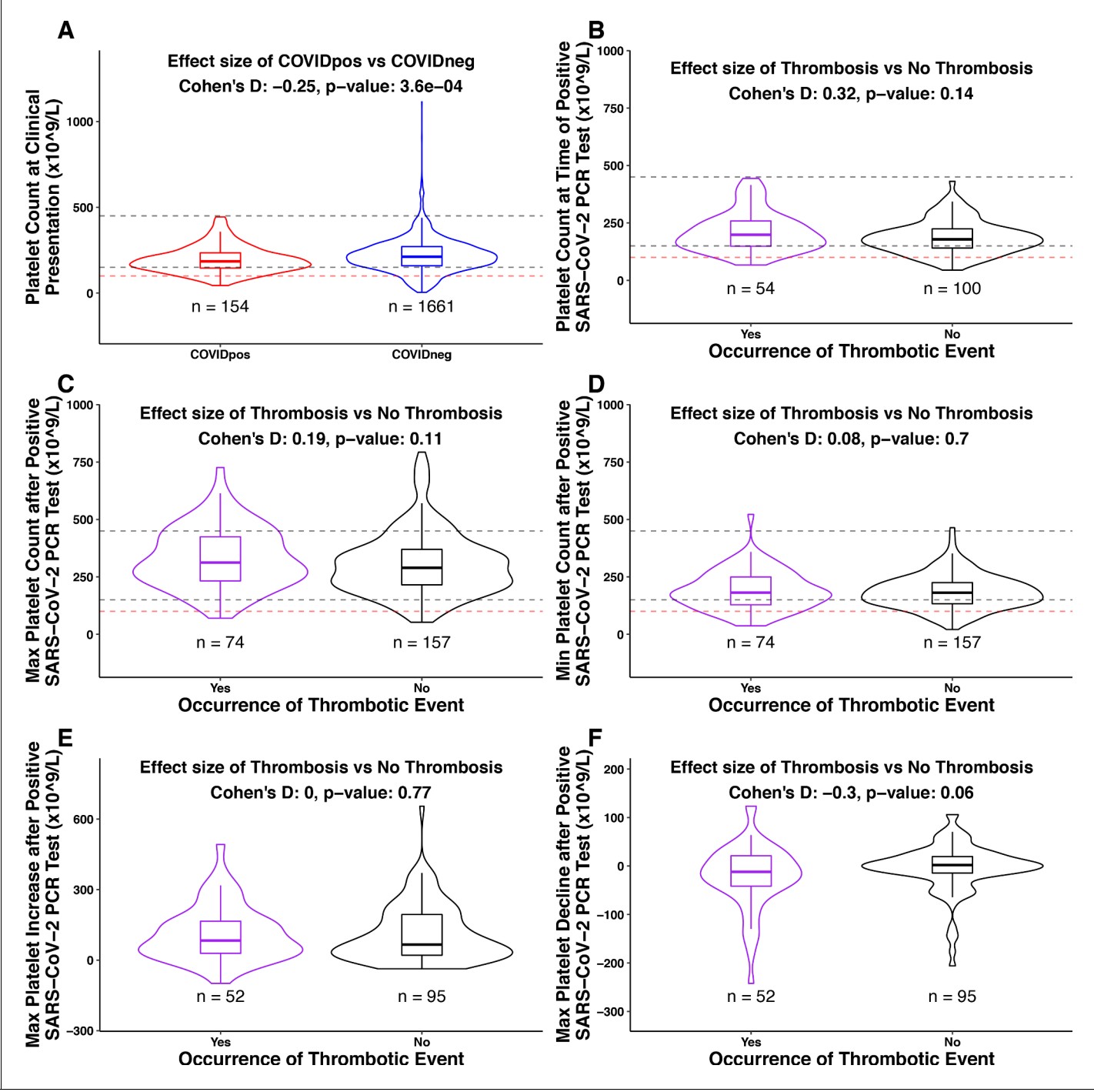

**Figure 5.** Association between platelet counts and thrombosis in the COVID_pos cohort. Box plots of platelet counts, min/max values, and maximum levels of increase/decline at specific time intervals for COVID_pos and COVID_neg cohorts and subgroups of the COVID_pos cohort with and without thrombotic events after SARS-CoV-2 diagnosis. In the subplot (A), we show platelet counts for COVID_pos (red) and COVID_neg (blue) cohorts. In subplots (B-F), we show platelet counts for COVID_pos patients who did and did not subsequently develop thromboses (purple and black, respectively). Horizontal dotted gray lines correspond to upper and lower limits of normal platelet counts (150–450 × 10^9/L), and horizontal red line shows 100 × 10^9/L. At the top of each plot, Cohen's D effect size and p-value from the Mann-Whitney statistical test are shown. (A) Platelet counts at the time of PCR testing for COVID_pos and COVID_neg cohorts. (B) Platelet counts at the time of PCR testing for COVID_pos patients who did and did not subsequently develop thromboses. (C) Maximum platelet counts (considering counts at and after positive PCR test date) for COVID_pos patients who did and did not subsequently develop thromboses. (D) Minimum platelet counts (considering counts at and after positive PCR test date) for COVID_pos patients who did and did not subsequently develop thromboses. (E) Maximum degree of platelet increases after positive PCR test date for COVID_pos patients who did

*Figure 5 continued on next page*

*Figure 5 continued*

and did not subsequently develop thromboses. (F) Maximum degree of platelet declines after positive PCR test date for COVID$_{pos}$ patients who did and did not subsequently develop thromboses.

## Consumptive coagulopathy contributes to only a small fraction of COVID-19 associated thromboses

The observed declining platelet counts and thrombocytopenia in the context of thrombosis in a small fraction of COVID$_{pos}$ patients are consistent with previous reports that fewer than 1% of survivors, but over 70% of non-survivors, meet the International Society on Thrombosis and Hemostasis (ISTH) criteria for disseminated intravascular coagulation (DIC; *Tang et al., 2020*). As was previously noted, hyperfibrinogenemia was among the strongest lab test features distinguishing COVID$_{pos}$ from COVID$_{neg}$ patients at diagnosis, but the subsequent downward trend (*Figure 3A*) could be attributed to a resolving acute phase response and/or consumption of fibrinogen in a systemic coagulopathy. Using our BERT-based sentiment extraction, we found that only five of the 2232 COVID$_{pos}$ patients that exhibited DIC-like symptoms, all of whom were included in our longitudinal cohort of 246 COVID$_{pos}$ patients (*Table 4*). Upon manual review of the EHR data for each patient, we found that two out of these five patients had confirmed diagnosis of DIC, while the remaining had high clinical suspicion and pending tests for DIC. This finding suggests that declining fibrinogen after COVID-19 diagnosis typically represents a physiologic return to normal range rather than pathologic coagulation factor consumption. To further examine the plasma fibrinogen trends among COVID-19 patients with DIC, with non-DIC thrombosis, and without thrombosis, we examined patient-level lab test trends from 10 individuals who were tested for fibrinogen both at the time of diagnosis and at least two times subsequently. The 10 patients for individual analysis were selected as the first 10 individuals with longitudinal fibrinogen lab testing data available.

This patient-level analysis indeed revealed multiple distinct trajectories with respect to fibrinogen and other coagulation parameters in COVID$_{pos}$ patients. Four of these ten individuals developed at least one blood clot during their hospital course. Only one was identified by our BERT model (and confirmed by manual EHR review) to have low-grade DIC, and as expected we found this patient's longitudinal lab test pattern to be consistent with consumptive coagulopathy (Patient 124; *Figure 6A*). At the time of diagnosis, this patient showed significant hyperfibrinogenemia with elevated D-dimers (1304.5 ng/mL) and a borderline normal platelet count ($153 \times 10^9$/L). Over the next 10 days, this patient's fibrinogen levels consistently decreased, reaching a minimum of 110 mg/dL

**Table 6.** Validation of the BERT model to identify the sentiment of thrombotic phenotypes in clinical notes.

Out-of-sample accuracy results of the BERT model to identify thrombotic phenotypes in 1000 randomly selected sentences from clinical notes which contained at least one mention of a thrombotic phenotype. The columns are (1) *Clotting phenotype:* thrombotic phenotype identified in the sentence, (2) *TP (true positives):* count of sentences in which the BERT model correctly identified the sentiment as 'Yes', (3) *TN (true negatives):* count of sentences in which the BERT model correctly identified the sentiment as not 'Yes', (4) *FP (false positives):* count of sentences in which the BERT model incorrectly identified the sentiment as 'Yes', (5) *FN: (false negatives):* count of sentences in which the BERT model incorrectly identified the sentiment as not 'Yes', (6) *Recall:* recall of the BERT model, equal to TP/(TP+FN), (7) *Precision:* precision of the BERT model, equal to TP/(TP+FP), (8) *Accuracy:* accuracy of the BERT model, equal to (TP+TN)/(TP+TN+FP+FN).

| Clotting phenotype | TP | TN | FP | FN | Recall | Precision | Accuracy |
|---|---|---|---|---|---|---|---|
| Deep vein thrombosis | 136 | 178 | 24 | 3 | 98% | 85% | 92% |
| Pulmonary embolism | 164 | 78 | 7 | 6 | 96% | 96% | 95% |
| Myocardial infarction | 212 | 65 | 3 | 3 | 99% | 99% | 98% |
| Venous thromboembolism | 3 | 97 | 7 | 0 | 100% | 30% | 93% |
| Thrombotic stroke | 5 | 0 | 0 | 0 | 100% | 100% | 100% |
| Cerebral venous thrombosis | 1 | 0 | 0 | 0 | 100% | 100% | 100% |
| Disseminated intravascular coagulation | 4 | 4 | 0 | 0 | 100% | 100% | 100% |
| Overall | 525 | 422 | 41 | 12 | 97.8% | 92.8% | 94.7% |

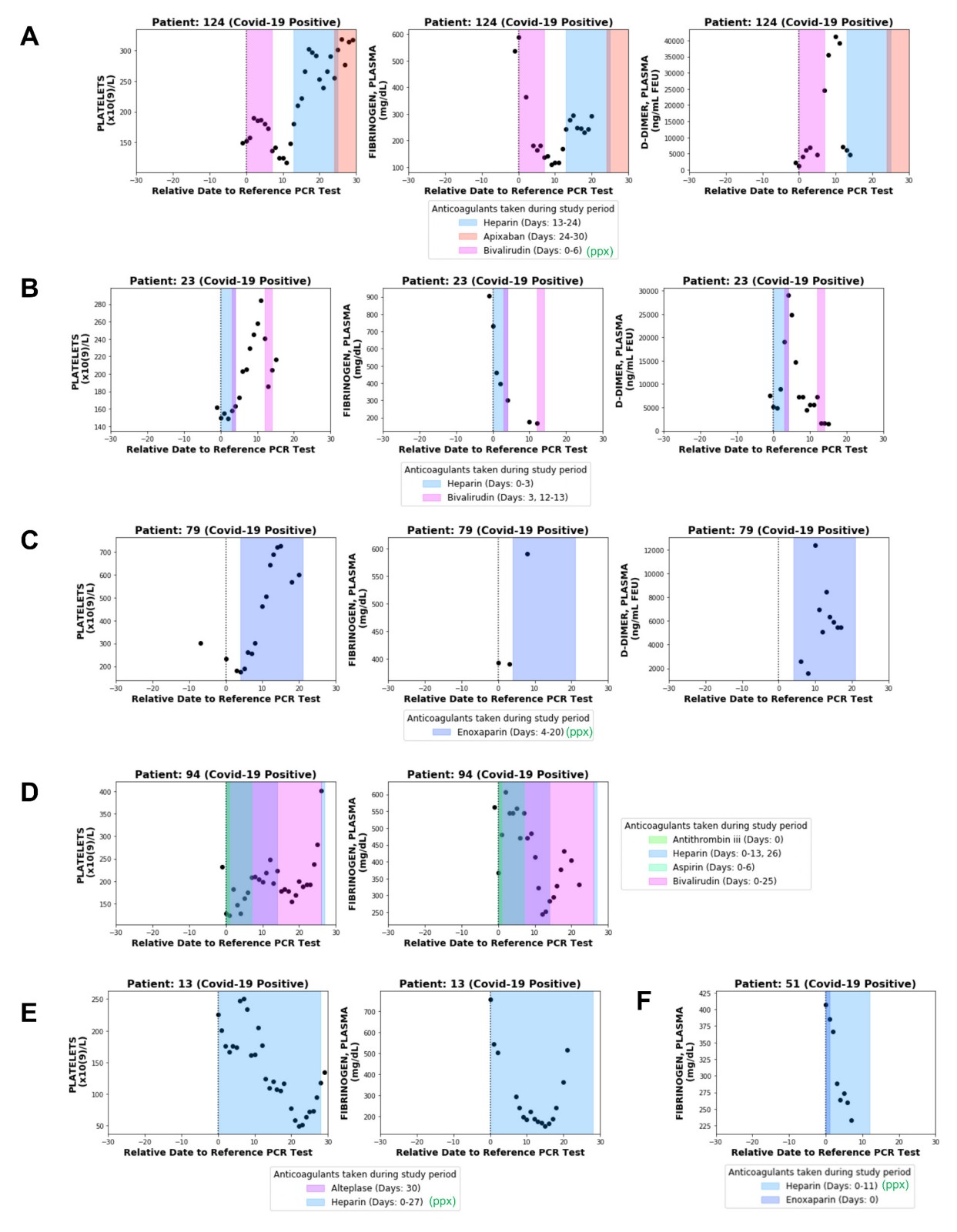

**Figure 6.** Longitudinal analyses of platelet counts, plasma fibrinogen, and D-dimer levels in individual patients with or without thrombotic disease. In each plot, shaded regions represent time periods when the patient was taking a specific anticoagulant or antiplatelet medication. Medications taken for prophylaxis are denoted in the legend with (ppx). (**A**) Patient 124 developed hemorrhagic and thrombotic phenotypes in the context of declining fibrinogen, declining platelets, and increasing D-dimers. This is consistent with a DIC-like coagulopathy. (**B**) Patient 23 developed clots in the setting of

*Figure 6 continued on next page*

*Figure 6 continued*

declining fibrinogen and elevated D-dimers but stable platelet counts which increased shortly thereafter. (C) Patient 79 developed clots while showing increases in platelet counts along with plasma fibrinogen and D-dimers. (D) Patient 94 developed clots with relatively stable platelet counts and steadily declining plasma fibrinogen. (E) Patient 13 did not develop clots or bleeding despite a coordinate decrease in platelet counts and fibrinogen which may be mistaken for a DIC-like coagulopathy. (F) Patient 51 did not develop clots despite showing a post-diagnosis decline in plasma fibrinogen similar to several patients in the thrombotic cohort.

on day 9. Similarly, after an initial recovery to $190 \times 10^9$/L the platelet counts consistently declined starting on day 2 post-diagnosis, reaching a minimum of $117 \times 10^9$/L on day 11. D-dimer levels exponentially increased after 5 days, reaching a maximum of 41,300 ng/mL on day 10. Phenotypically, this patient experienced both thrombotic (right internal jugular vein and right superior thyroid artery) and hemorrhagic (oropharyngeal and pulmonary) events. This combination of lab results and clinical manifestations is consistent with the diagnosis of DIC-like consumptive coagulopathy during the first week after COVID-19 diagnosis.

Lab test results from three other non-DIC thrombotic patients with longitudinal fibrinogen testing confirm the presence of alternative forms of coagulopathy in the COVID-19 population. Patient 23 developed a clot on day 4 post-diagnosis in the context of a declining fibrinogen level and increasing D-dimers but steady platelet counts, which actually increased shortly thereafter (*Figure 6B*). Patient 79 developed several clots after day 3 post-diagnosis in the setting of upward trending platelets (which eventually exceed the upper limit of normal) and elevated levels of both fibrinogen and D-dimers (*Figure 6C*). Patient 94 developed a clot on day 8 post-diagnosis with relatively stable platelet counts within normal limits and steadily declining fibrinogen levels (*Figure 6D*).

One hypothesis is that early elevations in plasma fibrinogen contribute to the clotting observed in the non-DIC like COVID$_{pos}$ cohort. This hypothesis may warrant further analysis in cohorts with more longitudinal fibrinogen data, but again it is important to note that several COVID$_{pos}$ patients who presented with hyperfibrinogenemia did not go on to develop thromboses (*Figure 6E–F*). This emphasizes that a steady post-diagnosis decline in plasma fibrinogen may represent physiologic resolution of the acute phase response rather than a pathologic consumption of fibrinogen and other coagulation factors (*Figure 6B,D–F*).

Taken together, this analysis affirms that a DIC-like coagulopathy resulting in a combination of hemorrhage and thrombosis can develop in the setting of COVID-19 infection. However, the observations that DIC was formally diagnosed in only five of 2232 COVID$_{pos}$ patients and emphasizes that consumptive coagulopathy is an exception rather than the rule as it pertains to thrombotic phenotypes in COVID-19 patients. These results should be considered as a preliminary characterization of COVID-associated coagulopathies (CAC) and will be updated as patient counts increase with the continued evolution of the COVID-19 pandemic.

## Discussion

Many studies on clinical characteristics and lab tests are shedding light on the spectrum of hematological parameters associated with COVID-19 patients. In an initial study of 41 patients from Wuhan, the blood counts in COVID$_{pos}$ patients showed leukopenia and lymphopenia, and prothrombin time and D-dimer levels were higher in ICU patients than in non-ICU patients (*Huang et al., 2020*). Another study based on 343 Wuhan COVID$_{pos}$ patients found that a D-dimer level of at least 2.0 μg/mL could predict mortality with a sensitivity of 92.3% and a specificity of 83.3% (*Zhang et al., 2020*). An independent study of 43 COVID-19 patients found significant differences between mild and severe cases in plasma interleukin-6 (IL-6), D-dimers, glucose, thrombin time, fibrinogen, and C-reactive protein ($p<0.05$; *Gao et al., 2020*). While such studies indeed highlight that hematological and inflammatory abnormalities are prevalent in COVID$_{pos}$, a high-resolution temporal understanding of how these parameters evolve in COVID-19 patients post diagnosis has not been established. Specifically, in the wake of accumulating evidence for hypercoagulability in COVID$_{pos}$ patients, there are important clinical questions emerging regarding the necessity of and guidelines for thromboprophylaxis in patient management.

DIC-like consumptive coagulopathy in COVID-19 has been a point of concern in severely ill COVID-19 patients. Particularly in patients with ARDS, multiple organ dysfunction syndrome (MODS)

**Table 7.** General characteristics of patients with SARS-CoV-2 PCR testing.

General demographic characteristics of all patients who underwent SARS-CoV-2 PCR testing in the Mayo Clinic EHR database from February 15, 2020 to May 28, 2020. Includes summary characteristics for: (A) all patients with at least one SARS-CoV-2 PCR test, and (B) patients with at least one SARS-CoV-2 PCR test and longitudinal testing data available (i.e. patient received the same lab test on 3 separate days within + / − 30 days of PCR testing date).

**(A) Demographics of all patients with PCR testing data**

|  | COVID$_{pos}$ | COVID$_{neg}$ |
|---|---|---|
| Total number of patients | 2232 | 72,354 |
| Gender: | | |
| Male | 1153 (52%) | 31,613 (44%) |
| Female | 1074 (48%) | 40,714 (56%) |
| Race: | | |
| White | 1115 (50%) | 62,605 (87%) |
| Black | 420 (19%) | 2792 (3.9%) |
| Asian | 151 (6.8%) | 1719 (2.4%) |
| American Indian | 29 (1.3%) | 302 (0.42%) |
| Other | 517 (23%) | 4936 (6.8%) |

**(B) Demographics of patients with PCR testing data and longitudinal testing data**

| Test | Units | Perturbation | Original time window | Count COVID$_{pos}$ | Count COVID$_{neg}$ | Mean COVID$_{pos}$ | Mean COVID$_{neg}$ | Cohen's D | BH-adjusted M-W p-value |
|---|---|---|---|---|---|---|---|---|---|
| Activated Partial Thrombopl Time, P | sec | −1 day | Days 7–9 Post-Dx | 26 | 72 | 50.1 | 38 | 0.57 | 0.034 |
| Activated Partial Thrombopl Time, P | sec | +1 day | Days 7–9 Post-Dx | 17 | 58 | 55 | 37.5 | 0.81 | 0.014 |
| Activated Partial Thrombopl Time, P | sec | −1 day | Days 10–12 Post-Dx | 16 | 57 | 56.9 | 38.4 | 0.808 | 9.10E-03 |
| Activated Partial Thrombopl Time, P | sec | +1 day | Days 10–12 Post-Dx | 15 | 60 | 56.9 | 38 | 1.106 | 2.60E-03 |
| Activated Partial Thrombopl Time, P | sec | −1 day | Days 13–15 Post-Dx | 15 | 52 | 55.5 | 37.8 | 1.041 | 0.014 |
| Activated Partial Thrombopl Time, P | sec | +1 day | Days 13–15 Post-Dx | 14 | 48 | 51.8 | 37.1 | 0.962 | 0.015 |
| Activated Partial Thrombopl Time, P | sec | −1 day | Days 16–30 Post-Dx | 22 | 156 | 55.2 | 37 | 0.913 | 5.70E-03 |
| Activated Partial Thrombopl Time, P | sec | +1 day | Days 16–30 Post-Dx | 19 | 139 | 56 | 38.2 | 0.725 | 3.80E-02 |
| Fibrinogen, P | mg/dL | −1 day | Clinical presentation | 25 | 92 | 584.9 | 370.7 | 1.067 | 1.20E-04 |
| Fibrinogen, P | mg/dL | +1 day | Clinical presentation | 37 | 292 | 488.2 | 326.2 | 0.885 | 8.80E-06 |
| Fibrinogen, P | mg/dL | −1 day | Days 1–3 Post-Dx | 41 | 381 | 494.5 | 318 | 1.023 | 3.90E-07 |
| Fibrinogen, P | mg/dL | +1 day | Days 1–3 Post-Dx | 21 | 244 | 420.3 | 312.2 | 0.616 | 7.90E-03 |
| Fibrinogen, P | mg/dL | −1 day | Days 4–6 Post-Dx | 27 | 156 | 432.2 | 336 | 0.495 | 0.045 |
| Fibrinogen, P | mg/dL | +1 day | Days 4–6 Post-Dx | 24 | 105 | 472.2 | 333.2 | 0.712 | 0.025 |
| Platelets | ×10(9)/L | −1 day | Pre-diagnosis | 34 | 575 | 187.3 | 225.6 | −0.357 | 0.057 |
| Platelets | ×10(9)/L | +1 day | Pre-diagnosis | 118 | 1533 | 201.3 | 234.4 | −0.328 | 7.30E-04 |

**Table 8.** Lab test data availability in patients with SARS-CoV-2 PCR testing.

Lab test data availability for all patients who underwent SARS-CoV-2 PCR testing in the Mayo Clinic EHR database from February 15, 2020 to May 28, 2020. Includes counts of lab tests and counts of patients with 1+ and 3+ lab tests both overall and for selected coagulation-related lab tests (activated partial thromboplastin time, D-dimer, fibrinogen, platelets, and prothrombin time).

| | COVID$_{pos}$ | COVID$_{neg}$ |
|---|---|---|
| Total number of patients | 2232 | 72,354 |
| Number of patients with 1+ lab test | 566 (25%) | 35,188 (49%) |
| Number patents with 1+ test from day −30 to day −1 | 299 (13%) | 23,116 (32%) |
| Number patents with 1+ test from day 0 to day 30 | 452 (20%) | 28,666 (40%) |
| Number of patients with 3+ lab tests of the same type | 246 (11%) | 13,666 (19%) |
| Total number of lab tests | 98,753 | 32,40,491 |
| Number of lab tests from day −30 to day −1 | 12,120 | 10,33,762 |
| Number of lab tests from day 0 to day 30 | 86,633 | 22,06,729 |
| ACTIVATED PTT | | |
| Number of lab tests | 362 | 6042 |
| Number of patients with 1+ lab test | 93 (4.0%) | 3544 (4.9%) |
| Number of patients with 3+ lab tests | 20 (0.86%) | 406 (0.56%) |
| D-DIMER, P | | |
| Number of lab tests | 911 | 2846 |
| Number of patients with 1+ lab test | 247 (11%) | 2395 (3.3%) |
| Number of patients with 3+ lab tests | 99 (4.4%) | 56 (0.077%) |
| FIBRINOGEN, P | | |
| Number of lab tests | 278 | 3,017 |
| Number of patients with 1+ lab test | 84 (3.8%) | 1217 (1.7%) |
| Number of patients with 3+ lab tests | 18 (0.81%) | 273 (0.38%) |
| PLATELETS | | |
| Number of lab tests | 2646 | 1,08,722 |
| Number of patients with 1+ lab test | 500 (22%) | 30,732 (42%) |
| Number of patients with 3+ lab tests | 231 (10%) | 11544 (16%) |
| PROTHROMBIN TIME, P | | |
| Number of lab tests | 711 | 28,007 |
| Number of patients with 1+ lab test | 197 (8.8%) | 10,446 (14%) |
| Number of patients with 3+ lab tests | 46 (2.1%) | 2502 (3.5%) |

is the predominant cause of death. A recent study suggested that DIC was associated with MODS during the early stage of ARDS and that persistent DIC may also have a role in this association (*Gando et al., 2020*). Our study focusing on COVID-19 patients with longitudinal lab data suggests that COVID-19 is indeed associated with modulation of coagulation related parameters such as platelet counts, fibrinogen levels, and clotting time (*Figure 2*). However, the majority of thrombotic events in COVID-19 patients with longitudinal lab testing are not the result of a DIC-like consumptive coagulopathy, as this only occurs in a small subset (*Table 4*).

The ability to derive this longitudinal understanding of COVID-19 progression, including laboratory abnormalities and their associated clinical manifestations, mandates the synthesis of structured and unstructured EHR data (e.g. lab tests and clinical notes) at a large scale. The fact that tens of thousands of patients have undergone SARS-CoV-2 testing at major academic medical centers (AMCs) provides an abundance of potential data to perform this analysis but also poses significant challenges from a practicality standpoint. Manual review and curation of patient trajectories and

**Table 9.** Lab test data availability in patients with SARS-CoV-2 PCR testing and longitudinal lab data.

Lab test data availability for all patients who underwent SARS-CoV-2 PCR testing in the Mayo Clinic EHR database from February 15, 2020 to May 28, 2020 with longitudinal testing data available (i.e. patient received the same lab test on three separate days within + / − 30 days of PCR testing date). Includes counts of lab tests and counts of patients with 1+ and 3+ lab tests both overall and for selected coagulation-related lab tests (activated partial thromboplastin time, D-dimer, fibrinogen, platelets, and prothrombin time).

| | $COVID_{pos}$ | $COVID_{neg}$ |
|---|---|---|
| Total number of patients | 246 | 13,666 |
| Number patents with 1+ test from day −30 to day −1 | 150 (61%) | 11,567 (85%) |
| Number patents with 1+ test from day 0 to day 30 | 240 (98%) | 13,501 (99%) |
| Total number of lab tests | 89,587 | 2,634,070 |
| Number of lab tests from day −30 to day −1 | 8698 | 763,808 |
| Number of lab tests from day 0 to day 30 | 80,889 | 1,870,262 |
| ACTIVATED PTT | | |
| Number of lab tests | 355 | 5186 |
| Number of patients with 1+ lab test | 86 (35%) | 2722 (20%) |
| Number of patients with 3+ lab tests | 20 (8.1%) | 406 (3.0%) |
| D-DIMER, P | | |
| Number of lab tests | 855 | 1720 |
| Number of patients with 1+ lab test | 197 (80%) | 1293 (9.5%) |
| Number of patients with 3+ lab tests | 99 (40%) | 56 (0.41%) |
| FIBRINOGEN, P | | |
| Number of lab tests | 275 | 2965 |
| Number of patients with 1+ lab test | 81 (33%) | 1168 (8.5%) |
| Number of patients with 3+ lab tests | 18 (7.3%) | 273 (2%) |
| PLATELETS | | |
| Number of lab tests | 2343 | 87,517 |
| Number of patients with 1+ lab test | 245 (100%) | 13,399 (98%) |
| Number of patients with 3+ lab tests | 231 (94%) | 11,544 (84%) |
| PROTHROMBIN TIME, P | | |
| Number of lab tests | 676 | 24,489 |
| Number of patients with 1+ lab test | 165 (67%) | 7209 (53%) |
| Number of patients with 3+ lab tests | 46 (19%) | 2502 (18%) |

associated testing results is not practical. It is not likely to provide comprehensive or even entirely accurate individual patient records. Rather, triangulation across datasets, including lab measurements, clinical notes, and prescription information, using a scalable digitized approach to extract structured data along with sentiment-surrounded clinical phenotypes and outcomes enables us to efficiently perform this analysis in a timely fashion.

By developing and deploying such a digitized platform on the entirety of EHR data from a large AMC, we have identified in an unbiased manner, laboratory test-based abnormalities that differentiate $COVID_{pos}$ patients from $COVID_{neg}$ patients. The abnormalities in coagulation-related tests, including fibrinogen and platelets, were intriguing in the context of literature reporting the occurrence of various clotting phenotypes in COVID-19 patients, including DIC-like consumptive coagulopathies along with more isolated clotting events in the lungs, central nervous system, and other tissues (*Tang et al., 2020*; *Klok et al., 2020*; *Levi et al., 2020*). Our finding that consumptive coagulopathy represents a minority of COVID-19 associated clotting events provides context for other

studies, which have reported overt DIC or DIC-like disease in over 70% of non-survivors but far lower fractions of survivors (*Tang et al., 2020*). As the pandemic continues to evolve and the patient counts increase over the coming months, we will be monitoring and reporting any updates to the clinical and laboratory observations drawn in this study.

Notwithstanding the preliminary nature of the analysis presented in this study, the results highlight that consumptive coagulopathy should be considered in the minority of COVID_pos patients with significant serial reductions in platelet counts. It remains to be seen whether the post-diagnosis platelet increases or early hyperfibrinogenemia which we observed may contribute mechanistically to the clotting in the much larger non-DIC thrombotic COVID-19 population. It is important to note that despite the trend of increasing platelets, the platelet count only extended above the normal range ($>450\times10^9$/L) after the PCR date in few COVID_pos patients with serial measurements, and the development of such outright thrombocytosis was observed with similar frequencies in the thrombotic and non-thrombotic cohorts (*Figure 5C*). Further, the fact that several patients with elevated fibrinogen (i.e. >400 mg/dL) at presentation did not develop thromboses suggests that early hyperfibrinogenemia is not a singular driver of subsequent clotting events, but a small sample size (n = 10 patients; nine non-thrombotic vs. one thrombotic) limited the power of this analysis (*Figure 6*).

Despite these caveats, this linking of longitudinal trends to patient outcomes provides several useful pieces of clinical information. First, hyperfibrinogenemia is to be expected in COVID-19 patients around the time of diagnostic testing. Furthermore, declining fibrinogen levels shortly after diagnosis are also expected and likely represent the resolution of acute phase response in most patients rather than a decline secondary to the onset of consumptive coagulation. In addition, borderline or overt thrombocytopenia is common in COVID-19 patients at the time of clinical presentation, and the initial platelet count does not robustly predict patients who are likely to develop thromboses. After diagnosis, COVID-19 patients generally show an upward trend in platelets. Patients whose platelets trend down after diagnosis should be monitored, as platelet reductions after clinical presentation are associated with thromboses and significant reductions may be indicative of ongoing consumptive coagulopathy.

One unavoidable limitation of this study is that we restrict our analysis to patients which have longitudinal lab testing data available. While the inclusion criteria is naturally biased, we consider this study population to be of high clinical interest because these patients are highly enriched for severe thrombotic events during the study period (see *Table 5*). Further, in the propensity score matching step of the analysis, we are able to construct a control cohort that is similar to the COVID_pos cohort in these enriched dimensions. To provide additional color on the distinctive attributes of the study population, we provide a summary of the clinical characteristics of the study population versus all patients with PCR tests during the same time period (see *Table 7*). In addition, we provide the median numbers of lab tests per patient for selected coagulation-related lab tests (fibrinogen, platelets, PTT, APTT, D-dimer) and total lab tests (*Tables 8* and *9*).

It is important to note that while we center the study period around the PCR testing date, this date may not correspond to the same disease state of COVID-19 for each individual in the COVID_pos cohort. To account for the potential variability in disease progression, we have performed a sensitivity analysis on the time intervals (*Table 3*). Additionally, there are several covariates that may influence these longitudinal trends and should be explored further. For example, we have already considered whether previous or concomitant administration of anticoagulants or antiplatelet agents influences patient lab test results and/or outcomes. Similarly, in the future, we intend to explore whether longitudinal lab measurement trends differ between outpatient, inpatient, and ICU admitted patient cohorts. New datasets can also be utilized; for example, rather than grouping patients by the identified thromboembolic phenotypes extracted from the clinical notes alone, patients could be stratified by those who had imaging studies (duplex ultrasound, CT scan, etc.) performed, and phenotypes could be directly extracted from these procedural reports. As more data accumulates from COVID_pos and COVID_neg patients in the coming months, these analyses need to be expanded to assess similarities and differences in the temporal trends of laboratory test results among a wider range of patient subgroups relevant for COVID-19 outcomes, such as those who have pre-existing conditions (e.g. diabetes, hypertension, obesity, malignancies) or patients who are on specific medication (e.g. ACE inhibitors, statins, immunosuppressants).

In summary, this work demonstrates significant progress toward enabling scaled and digitized analyses of longitudinal unstructured and structured EHRs to identify variables (e.g. laboratory

results) which are associated with relevant clinical phenotypes (e.g. COVID-19 diagnosis and outcomes). In doing so, we identified trends in lab test results which may be relevant to monitor in COVID-19 patients and warrant both clinical and mechanistic follow-up in more targeted and explicitly controlled prospective analyses.

## Materials and methods

### Study design, setting and patient population

This is a retrospective study of patients who underwent polymerase chain reaction (PCR) testing for suspected SARS-CoV-2 infection at the Mayo Clinic and hospitals affiliated to the Mayo health system. This research was conducted under IRB 20–003278, 'Study of COVID-19 patient characteristics with augmented curation of Electronic Health Records (EHR) to inform strategic and operational decisions'. For further information regarding the Mayo Clinic Institutional Review Board (IRB) policy, and its institutional commitment, membership requirements, review of research, informed consent, recruitment, vulnerable population protection, biologics, and confidentiality policy, please refer to www.mayo.edu/research/institutional-review-board/overview.

### Longitudinal lab testing tied to COVID-19 PCR diagnostic testing

We analyzed data from 74,586 patients who received PCR tests from the Mayo Clinic between February 15, 2020 to May 28, 2020. Among this population, 2232 patients had at least one positive SARS-CoV-2 PCR test result, and 72,354 patients had all negative PCR test results. In order to align the data for the analysis of aggregate longitudinal trends, we selected a reference date for each patient. For patients in the $COVID_{pos}$ cohort, we used the date of the first positive PCR test result as the reference date (day = 0). For patients with all negative PCR tests, we used the date of the first PCR test result as the reference date (day = 0). We defined the study period for each patient to be 30 days before and after the PCR testing date. Patients with contradictory PCR test results were excluded for the purpose of this analysis; for example, a positive PCR test result and a negative PCR test result on the same day, or a positive PCR test result followed immediately by several negative PCR test results.

Over 4 million test results from 6298 different types of lab tests were recorded for the patients who received PCR tests in the 60-day window surrounding their PCR testing dates at the Mayo Clinic campuses in Minnesota, Arizona, and Florida. Among these lab tests, we restricted our analysis to 194 tests with at least 1000 observations total and at least 10 observations from the $COVID_{pos}$ cohort among the patients with PCR testing on or before May 8, 2020. In addition, we considered different subsets of the $COVID_{pos}$ cohort for the analysis of each of the 194 lab tests, due to differences in availability of testing results. For each lab test, we consider the results from patients with three or more observations during the study period.

In the end, there are 246 SARS-COV-2 positive and 13,666 SARS-CoV-2 negative patients that had three or more test results during the study period for at least one of the assays among the 194 lab tests considered. We take this set of 246 COVID-19 positive patients to be the $COVID_{pos}$ cohort. In order to construct the $COVID_{neg}$ cohort from the 13,666 COVID-19 negative patients, we apply propensity score matching, which is described in the next section.

### Propensity score matching to select the final $COVID_{neg}$ cohort

To construct a $COVID_{neg}$ cohort similar in baseline clinical covariates to the $COVID_{pos}$ cohort, we employ 1:10 propensity score matching (*Austin, 2011*). In particular, first we trained a regularized logistic regression model to predict the likelihood that each patient will have a positive or negative COVID-19 test result, using the following covariates: demographics (age, gender, race), anticoagulant/antiplatelet medication use (orders for alteplase, antithrombin III, apixaban, argatroban, aspirin, bivalirudin, clopidogrel, dabigatran, dalteparin, enoxaparin, eptifibatide, heparin, rivaroxaban, warfarin in the past year and in the past 30 days), pre-existing coagulopathies (medical history of thrombotic phenotypes including: deep vein thrombosis, pulmonary embolism, myocardial infarction, venous thromboembolism, thrombotic stroke, cerebral venous thrombosis, and disseminated intravascular coagulation from day −365 to day −31 relative to the PCR testing date), and hospitalization status (i.e. whether or not the patient was hospitalized within the past 30 days of PCR testing).

Using the predictions from the logistic regression model as propensity scores, we then matched each of the 246 patients in the COVID$_{pos}$ cohort to 10 patients out of the 13,666 COVID-19 negative patients, using greedy nearest-neighbor matching without replacement (*Austin, 2011*; *Austin, 2014*). As a result, we ended up with a final COVID$_{neg}$ cohort that included 2460 patients with similar baseline characteristics to the COVID$_{pos}$ cohort. The characteristics of the two cohorts are summarized in *Table 1*.

Further, for the analyses conducted on individual lab tests, which include only a subset of patients from the COVID$_{pos}$ cohort, we use the propensity scores to match each patient from the COVID$_{pos}$ cohort to 10 patients from the COVID$_{neg}$ cohort which have the most similar propensity scores and lab tests available. For example, for the fibrinogen lab test, in which we have data on 81 patients from the COVID$_{pos}$ cohort, we select 810 patients from the COVID$_{neg}$ cohort and the most similar propensity scores to be the control group. In this way, we ensure that all of the comparisons are done between subsets of the positive and negative cohorts with similar propensity scores, and therefore similar underlying characteristics.

## Statistical significance assessments for lab test differences over prognostic time intervals for SARS-CoV-2 infection

We conduct a systematic statistical analysis to identify tests that show significant differentiation among the COVID$_{pos}$ cohort during a set of predetermined prognostic time intervals for SARS-CoV-2 infection. In particular, we group the lab test measurements for each patient into the following nine time intervals relative to their date of PCR testing: pre-infection (days −30 to −11), pre-PCR (days −10 to −2), time of clinical presentation (days −1 to 0), and post-PCR phases 1 (days 1 to 3), 2 (days 4 to 6), 3 (days 7 to 9), 4 (days 10 to 12), 5 (days 13 to 15), and 6 (days 16 to 30).

For each lab test and for each of each of our nine pre-specified time intervals, we compared the mean lab test value among patients who underwent at least one such lab test in the COVID$_{pos}$ cohort over that time interval to the mean lab test value in the COVID$_{neg}$ (matched) cohort over that time window. We only considered (lab test, time interval) pairs in which there were at least three patients contributing to laboratory test results in both groups. Specifically, for each (lab test, time interval) pair, we conducted the following procedure:

1. Compute (patient, time interval) averages: We compute the average lab test values for each patient in the COVID$_{pos}$ and COVID$_{neg}$ (matched) cohorts during the specified time interval.
2. Statistical hypothesis testing: We conduct a Mann-Whitney $U$ test in order to test the null hypothesis that the average lab test results for each of the (patient, time interval) pairs from the COVID$_{pos}$ and COVID$_{neg}$ (matched) cohorts come from the same distribution. In addition, we compute the Cohen's D statistic as a measure of the effect size.

Once we have the statistics and p-values for each (test, time window) pair, in order to account for multiple hypotheses, we apply the Benjamini-Hochberg (BH) procedure with FDR controlled at 0.05. The results from the systematic comparisons which met our thresholds for effect size and statistical significance (Cohen's D > 0.35, BH-adjusted Mann-Whitney p-value <0.05) are shown in *Table 2*.

## Sensitivity analysis to assess the impact of perturbed clinical time windows

We perform a sensitivity analysis to assess whether or not the key findings from the systematic statistical assessment remain the same if we perturb the considered time intervals. In particular, we repeat the statistical analysis with the time intervals shifted forward or backward 1 day for all patients. For the forward shifted sensitivity analysis, the new time intervals under consideration are: pre-infection (days −30 to −10), pre-PCR (days −9 to −1), time of clinical presentation (days 0 to 1), and post-PCR phases 1 (days 2 to 4), 2 (days 5 to 7), 3 (days 8 to 10), 4 (days 11 to 13), 5 (days 14 to 16), and 6 (days 17 to 30). For the backward shifted sensitivity analysis, the new time intervals under consideration are: pre-infection (days −30 to −12), pre-PCR (days −11 to −3), time of clinical presentation (days −2 to −1), and post-PCR phases 1 (days 0 to 2), 2 (days 3 to 5), 3 (days 6 to 8), 4 (days 9 to 11), 5 (days 12 to 14), and 6 (days 15 to 30). For both the forward and backward sensitivity analyses, we apply the same thresholds of effect size and significance (Cohen's D > 0.35, BH-adjusted Mann-Whitney p-value <0.05), and we compare the results to the original time intervals.

From this analysis, we observe consistent results (i.e. comparisons meeting same criteria of significance and effect) on (i) both perturbations in 83 out of 130 (64%) lab test trends identified in *Table 2* and (ii) at least one perturbation in 114 of 130 (87%) lab test trends. In *Table 3*, we report the specific results of the time shifted windows for five coagulation-related lab tests (fibrinogen, platelets, prothrombin time, activated partial thromboplastin time, and D-dimer).

### Augmented curation of anticoagulant administration and the coagulopathy outcomes from the unstructured clinical notes and their triangulation to structured EHR databases

A state-of-the-art BERT-based neural network (*Devlin et al., 2018*) was previously developed to classify sentiment regarding a diagnosis in the EHR (*Wagner et al., 2020*). Sentences containing phenotypes were classified into the following categories: Yes (confirmed diagnosis), No (ruled out diagnosis), Maybe (possibility of disease), and Other (alternate context, e.g. family history of disease). The neural network used to perform this classification was trained using nearly 250 different phenotypes and 18,500 sentences and achieves 93.6% overall accuracy and over 95% precision and recall for Yes/No sentiment classification (*Wagner et al., 2020*). Here, this model was used to classify the sentiment around coagulopathies in the unstructured text of the 246 $COVID_{pos}$ and 13,666 $COVID_{neg}$ patients' clinical notes, structuring this information so that it could be compiled with longitudinal lab measurement and medication information.

In particular, we used the BERT model to identify the seven coagulopathy phenotypes mentioned in clinical notes in the Mayo Clinic EHR database, including: deep vein thrombosis, pulmonary embolism, myocardial infarction, venous thromboembolism, thrombotic stroke, cerebral venous thrombosis, and disseminated intravascular coagulation. We validated the performance of this model for these phenotypes on a set of 1000 randomly selected sentences from the clinical notes of the patients in the study population. In *Table 6*, we report the out-of-sample accuracy metrics for the BERT model on this set of sentences, using manually curated labels provided by one of the study's authors (CP) to be the ground truth. We demonstrate that the model performs well in the task of identifying thrombotic phenotypes in clinical notes, with an overall accuracy of 94.7%, recall of 97.8%, and precision of 92.8%.

## Acknowledgements

The authors thank Mathai Mammen, James List, JoAnne Foody, Patrick Lenehan, Murali Aravamudan, Rakesh Barve, Sankar Ardhanari, and Vishy Thiagarajan, for their helpful feedback.

## Additional information

### Competing interests

Colin Pawlowski, Tyler Wagner, Arjun Puranik, Karthik Murugadoss, Liam Loscalzo, AJ Venkatakrishnan: This author is an employee of nference with financial interests in the company. Rajiv K Pruthi, Damon E Houghton, John C O'Horo, Amy W Williams, Gregory J Gores, John Halamka, Andrew D Badley: One or more of the investigators associated with this project and Mayo Clinic have a Financial Conflict of Interest in technology used in the research and that the investigator(s) and Mayo Clinic may stand to gain financially from the successful outcome of the research. Mayo Clinic is an investor in nference. This research has been reviewed by the Mayo Clinic Conflict of Interest Review Board and is being conducted in compliance with Mayo Clinic Conflict of Interest policies. William G Morice II: One or more of the investigators associated with this project and Mayo Clinic have a Financial Conflict of Interest in technology used in the research and that the investigator(s) and Mayo Clinic may stand to gain financially from the successful outcome of the research. This research has been reviewed by the Mayo Clinic Conflict of Interest Review Board and is being conducted in compliance with Mayo Clinic Conflict of Interest policies. The author is also involved in the Mayo Clinic Laboratories. Elliot S Barnathan, Hideo Makimura, Najat Khan: This author is an employee of the Janssen pharmaceutical companies of J&J with financial interests in the company. Venky Soundararajan: The author is an employee of nference and has financial interests in the company. Outside the submitted work, Venky Soundararajan is listed as inventor of the following patent: "Systems,

methods, and computer readable media for visualization of semantic information and inference of temporal signals indicating salient associations between life science entities" (US20180082197A1).

## Funding
No external funding was received for this work.

## Author contributions
Colin Pawlowski, Formal analysis, Validation, Investigation, Methodology, Writing - original draft, Writing - review and editing; Tyler Wagner, Formal analysis, Supervision, Validation, Investigation, Methodology, Writing - original draft, Project administration, Writing - review and editing; Arjun Puranik, Software, Formal analysis, Validation, Investigation, Visualization, Methodology, Writing - review and editing; Karthik Murugadoss, Software, Formal analysis, Validation, Investigation, Methodology, Writing - review and editing; Liam Loscalzo, Formal analysis, Supervision, Investigation, Methodology, Project administration; AJ Venkatakrishnan, Formal analysis, Validation, Investigation, Methodology, Writing - review and editing; Rajiv K Pruthi, Supervision, Validation, Investigation, Methodology, Writing - original draft, Project administration, Writing - review and editing; Damon E Houghton, Formal analysis, Supervision, Validation, Investigation, Methodology, Project administration, Writing - review and editing; John C O'Horo, Amy W Williams, Gregory J Gores, Supervision, Investigation, Project administration, Writing - review and editing; William G Morice II, Supervision, Investigation, Methodology, Project administration; John Halamka, Supervision, Project administration; Andrew D Badley, Supervision, Validation, Project administration, Writing - review and editing; Elliot S Barnathan, Hideo Makimura, Supervision, Validation, Investigation, Methodology, Writing - review and editing; Najat Khan, Conceptualization, Resources, Formal analysis, Supervision, Validation, Investigation, Methodology, Writing - review and editing; Venky Soundararajan, Conceptualization, Resources, Formal analysis, Supervision, Funding acquisition, Validation, Investigation, Methodology, Writing - original draft, Writing - review and editing

## Author ORCIDs
Colin Pawlowski (iD) https://orcid.org/0000-0003-2781-7507
Venky Soundararajan (iD) https://orcid.org/0000-0001-7434-9211

## Ethics
Human subjects: This research was conducted under IRB 20-003278, "Study of COVID-19 patient characteristics with augmented curation of Electronic Health Records (EHR) to inform strategic and operational decisions". All analysis of EHRs was performed in the privacy-preserving environment secured and controlled by the Mayo Clinic. nference, the Mayo Clinic, and the Janssen pharmaceutical companies of Johnson & Johnson (J&J) subscribe to the basic ethical principles underlying the conduct of research involving human subjects as set forth in the Belmont Report and strictly ensure compliance with the Common Rule in the Code of Federal Regulations (45 CFR 46) on the Protection of Human Subjects.

## Decision letter and Author response
Decision letter https://doi.org/10.7554/eLife.59209.sa1
Author response https://doi.org/10.7554/eLife.59209.sa2

# Additional files

## Supplementary files
• Transparent reporting form

## Data availability
De-identified data will be made available upon reasonable request to the corresponding author (Venky Soundararajan, venky@nference.net).

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
