## [Decision Letter]

**Acceptance summary:**

The paper analyzes a large EHR-based dataset of coagulation tests in COVID-19 patients to obtain an understanding of the kinetics of COVID-19 associated coagulopathy. By using machine learning to extract patterns data are provided which support that the majority of thrombotic events in COVID-19 patients are not the result of a DIC-like consumptive coagulopathy, and that this only occurs in a small subset.

**Decision letter after peer review:**

Thank you for submitting your article "Longitudinal laboratory testing tied to PCR diagnostics in COVID-19 patients reveals temporal evolution of coagulopathy" for consideration by *eLife*. Your article has been reviewed by four peer reviewers, and the evaluation has been overseen by a Reviewing Editor and Jos van der Meer as the Senior Editor. The following individuals involved in review of your submission have agreed to reveal their identity: Coen Maas (Reviewer #1); Jinbo Chen (Reviewer #4).

The reviewers have discussed the reviews with one another and the Reviewing Editor has drafted this decision to help you prepare a revised submission.

Summary:

The authors analyze a large EHR-based dataset of coagulation tests in COVID-19 patients to obtain an understanding of the kinetics of COVID-19 associated coagulopathy. It is a clear approach and the data are important for the clinics. They used machine learning to extract patterns. Overall the conclusions were drawn in many directions. Nonetheless, the paper asks a potentially important question and is adequately written and presented. The data support that the majority of thrombotic events in COVID-19 patients are not the result of a DIC-like consumptive coagulopathy, this only occurs in a small subset. This should also be the focus of the Discussion.

Essential revisions:

Patient characteristics:

– Please clinically define "pre-existing coagulopathies".

– It is a unclear what the comparator cohort adds: In part it is not clear who these patients are. Perhaps the controls should be age/sex-matched patients with other causes of hypoxemic respiratory failure/pneumonia? Or different COVID-19 disease severity subgroups? Otherwise what are we inferring from comparing these two groups?

– Ascertained rates of venous thromboembolism are much lower than what others have observed (e.g., Helms ICM 2020, Middeldorp pre-prints, etc.). Could this relate to ascertainment bias in this cohort? Perhaps a push not to perform diagnostic imaging studies lead to lower objective identification of venous thromboembolism than has been generally described in COVID-19?

– Patients are identified as COVID-19 positive or negative based on PCR testing. Actually, using the PCR test a positive/negative infection by SARS-CoV-2 is assessed; COVID-19 is the disease that can follow upon infection. This distinction should be made clear, as not all patients tested positive for SARS-CoV-2 infection necessarily develop COVID-19. What was the guideline followed for PCR-testing: displaying COVID-19 symptoms/ contact with infected persons/ other? This should also be explained and included.

– Introduction: "… straddling the date of the PCR test.…" Do the authors refer to the first PCR test here, as it is mentioned that multiple PCR tests may have been performed? Please include a definition.

Analysis:

– The analysis was restricted to patients who had serial (>/= 3) tests done. This could have led to a survival bias which should be acknowledged.

– This restriction also led to the exclusion of the vast majority of the cohort: 1,192 -> 181 COVID-19_pos_ patients. Why was there such a strong emphasis on longitudinal markers?

– The potential significance of these longitudinal trends are assessed with these tests as far as I can understand, which both only looks to see whether mean/median change was different between the groups, but moreover does not allow for adjustment. Why not use multilevel regression mixture models, longitudinal regression, etc?

– One of the three covariates used in the regularized logistic regression model to predict the likelihood of a positive infection (subsection “Propensity score matching to select the final COVID_neg_ cohort”) is anticoagulant/antiplatelet medication use. Is this covariate positively or negatively correlated to infection and what is the rationale for this? Table 1 comprises more details; however, these add to further confusion. The Table 1 legend states "anticoagulant/antiplatelet use within 30 days/1 year of PCR testing date" i.e. after the PCR test, while the table itself mentions "medication use in the preceding 30 days/1yr", so before the PCR test. Please clarify and adapt.

Figures:

– Figure 2A: Why are no data points provided for the -30 to 0 days of the COVID negative patients while these data are shown in Figure 2B and E?

– The same question as above but then for the APTT and D-dimer data in Figure 2 vs. Figure 3 and magnesium in Figure 1 vs. Figure 3. I guess the cohorts are different between the two figures, maybe this should be stated in the figure legends.

– Figure 3: we agree with the authors that the fibrinogen decline and platelet increase in COVID positive patients is re-emphasized in this manner. Also the increase in magnesium and decrease in alkaline phosphatase seem to stand out. Could the authors comment on this?

– Could the authors comment on the number of thrombotic events that were radiographically-confirmed?

– Figure 5: for the individual patients could the authors comment on the heparin therapy with regard to the source of heparin (LMWH vs. UFH) and dosing (prophylactic vs. therapeutic)?

– Figure 6 is very hard to understand; please find another way to graphically display the findings in a clear manner.

Specific statistical comments:

1) Cohort identification and description. (a) For the 1.3 million lab tests on 194 assays over the 60-day window: provide the mean/range of number of tests separately for positive and negative patients, and also provide the mean/range of the number of tests for pre- vs. post-index "0" date; (b) For positive patients, the proportion of those whose PCR positive test is the first test in the 60-day window; (c) If at all possible, COVID-19 related info at the first PCR test for all patients; (4) If possible, reasons for hospitalizations at the day=0;

2) It is not clear why it is important to assess lab tests in relation to diagnosis. For studying association between lab tests with diagnosis, ideally, the positive cases should use the "time of infection" as time zero which of course is intractable. But before variation in time from infection to diagnosis by PCR tests, the relevance of test results for diagnosis is unclear. Further, because of the propensity score matching, the cohort may not be suitable for assess the diagnosis: if some matching variables are associated with any of the test results, matching will artificially deflate the association. We therefore suggest the focus of this paper on prognosis only. The Abstract indeed focused on the prognosis, but diagnosis was mentioned multiple times in the manuscript.

3) BERT method: The accuracy of BERT was established in an unpublished manuscript developed by the same research group. What is the proportion of patients who were classified as "Maybe", "Yes" AND "No"? It is helpful if chart review is performed on 100, say, patients, to validate this algorithm in the study context;

4) Tables 3 and 4 indicate that the endpoints were significantly enriched for positive cases who had longitudinal data. But it is possible that patients received more tests because of indications of more negative outcomes. As a result, patients with longitudinal lab tests are representative of all positive patients as indicated by the enrichment. This is an important weakness of the study as this may be a source of bias, making the study results not generalizable. It is important to provide additional information on the comparing characteristics of patients with and without longitudinal tests. It may be worthwhile to provide information on reasons of repeating tests for negative patients.

---

## [Author Response]

Summary:The authors analyze a large EHR-based dataset of coagulation tests in COVID-19 patients to obtain an understanding of the kinetics of COVID-19 associated coagulopathy. It is a clear approach and the data are important for the clinics. They used machine learning to extract patterns. Overall the conclusions were drawn in many directions. Nonetheless, the paper asks a potentially important question and is adequately written and presented. The data support that the majority of thrombotic events in COVID-19 patients are not the result of a DIC-like consumptive coagulopathy, this only occurs in a small subset. This should also be the focus of the Discussion.

We thank the editors for their advice to focus the key conclusion of the paper. Accordingly, we have revised the manuscript’s Discussion section to reflect this suggestion. We find that the updated data for this manuscript also support this conclusion that DIC-like consumptive coagulopathy is relatively rare among COVID-19 patients, as shown in updated Table 3.

Essential revisions:Patient characteristics:– Please clinically define "pre-existing coagulopathies".

We consider the following phenotypes to be coagulopathies: deep vein thrombosis, pulmonary embolism, myocardial infarction, venous thromboembolism, thrombotic stroke, cerebral venous thrombosis, and disseminated intravascular coagulation. We have added a definition of pre-existing coagulopathies to the Introduction, which includes all patients who have at least one of the phenotypes from days -365 to -31 relative to the PCR testing date. In Table 1 of the revised manuscript, we have now added in the medical history of thrombotic events for the patients in the COVID-19 positive, negative, and negative (matched) cohorts.

– It is a unclear what the comparator cohort adds: In part it is not clear who these patients are. Perhaps the controls should be age/sex-matched patients with other causes of hypoxemic respiratory failure/pneumonia? Or different COVID-19 disease severity subgroups? Otherwise what are we inferring from comparing these two groups?

The comparator cohort was selected to match the COVID_pos_ cohort on a variety of clinical characteristics which may be confounding variables for thrombotic events and related lab tests that we observe during the study period. The clinical covariates that we controlled for initially included demographics (age, gender, race), hospitalization status, and anticoagulant/antiplatelet use, which can be viewed as a proxy variable for medical history of thrombotic events. In this revision, we have explicitly added in medical history of thrombotic events as clinical covariates to balance on, as shown in the updated Table 1 of the revised manuscript. We have added a few statements describing the updated COVID_neg_ (matched) cohort to the last paragraph of the Introduction.

– Ascertained rates of venous thromboembolism are much lower than what others have observed (e.g., Helms ICM 2020, Middeldorp pre-prints, etc.). Could this relate to ascertainment bias in this cohort? Perhaps a push not to perform diagnostic imaging studies lead to lower objective identification of venous thromboembolism than has been generally described in COVID-19?

Our study presents, for the first time, fine grained temporal resolution into lab results change in all COVID_pos_ patients. In particular, our study is not restricted to severe MODS or ARDS patients, as has been the focus of previous studies such as Helms ICM 2020, Middeldorp preprints, or https://pubmed.ncbi.nlm.nih.gov/32353745/ etc. We expect that this is the reason for the differences in venous thromboembolism rate reported across the different studies.

– Patients are identified as COVID-19 positive or negative based on PCR testing. Actually, using the PCR test a positive/negative infection by SARS-CoV-2 is assessed; COVID-19 is the disease that can follow upon infection. This distinction should be made clear, as not all patients tested positive for SARS-CoV-2 infection necessarily develop COVID-19. What was the guideline followed for PCR-testing: displaying COVID-19 symptoms/ contact with infected persons/ other? This should also be explained and included.

As recommended by the reviewers, the distinction between SARS-CoV-2 infection and the COVID-19 disease onset and progression has been clarified in the revised paper. The guidelines followed for PCR-testing included all of the following – displaying COVID-19 symptoms, contact with infected persons, and other factors such as potential work-place/community exposure as well as underlying conditions predisposing to SARS-CoV-2 infection. This has been noted in the second paragraph of the Introduction section of the manuscript.

– Introduction: "… straddling the date of the PCR test.…" Do the authors refer to the first PCR test here, as it is mentioned that multiple PCR tests may have been performed? Please include a definition.

We have added a statement to the Introduction, clarifying this point. For the COVID_pos_ cohort, we center the two month observation period around the date of the first positive PCR test for SARS-CoV-2, and for the COVID_neg_ cohort, we center the two month observation period around the date of the first PCR test for SARS-CoV-2.

Analysis:– The analysis was restricted to patients who had serial (>/= 3) tests done. This could have led to a survival bias which should be acknowledged.

We thank the reviewers for raising this point. We have added in a statement on potential survival bias to the Discussion section. In particular, we now provide a summary of the clinical characteristics of the study population vs. all patients with PCR tests during the same time period in Table 6. We have also included an additional table characterizing the key lab tests and their availability for the study population vs. all patients (see Tables 7-8).

– This restriction also led to the exclusion of the vast majority of the cohort: 1,192 -> 181 COVID-19_pos_ patients. Why was there such a strong emphasis on longitudinal markers?

The need for longitudinal data on the testing results, while constraining the study population size greatly, enables us to provide a fine-grained temporal resolution of COVID-associated coagulopathy for the first time. In addition, we consider this study population to be of high clinical interest because these patients are highly enriched for severe thrombotic events during the study period (see Table 4). We have added this in as a point in the seventh paragraph of the Discussion section.

– The potential significance of these longitudinal trends are assessed with these tests as far as I can understand, which both only looks to see whether mean/median change was different between the groups, but moreover does not allow for adjustment. Why not use multilevel regression mixture models, longitudinal regression, etc?

Thank you for the suggestion. We considered logistic regression for this analysis, to predict the presence/absence of thrombotic events using the longitudinal lab testing information available for each patient prior to the PCR testing date Day 0. However, due to large heterogeneity in lab tests conducted for the study population, and the relative sparsity of lab testing data available for the COVID_pos_ population prior to Day 0, we were unable to train an accurate predictive model using the available data. However, we expect that as additional data becomes available this is a promising area for future research, and we have added a note on this in the Discussion section.

– One of the three covariates used in the regularized logistic regression model to predict the likelihood of a positive infection (subsection “Propensity score matching to select the final COVID_neg_ cohort”) is anticoagulant/antiplatelet medication use. Is this covariate positively or negatively correlated to infection and what is the rationale for this? Table 1 comprises more details; however, these add to further confusion. The Table 1 legend states "anticoagulant/antiplatelet use within 30 days/1 year of PCR testing date" i.e. after the PCR test, while the table itself mentions "medication use in the preceding 30 days/1yr", so before the PCR test. Please clarify and adapt.

We observe that anticoagulant/antiplatelet use is negatively correlated with SARS-CoV-2 infection (see Table 1). We hypothesize that the reason for this is that patients who are on anticoagulants/antiplatelets generally have more underlying conditions and thus take more precautionary measures to avoid SARS-CoV-2 infection. In addition, Table 1 shows anticoagulant/antiplatelet use in the 30 days/1 year prior to the PCR testing date. We have revised the name of this covariate in Table 1 for clarification.

Figures:– Figure 2A: Why are no data points provided for the -30 to 0 days of the COVID negative patients while these data are shown in Figure 2B and E?

In the revised manuscript, Figure 2A has been renamed to Figure 3A. This plot, which shows the temporal trends of the Fibrinogen lab tests, has no data points between days -30 to 0 because none of the patients in the COVID_pos_ cohort had Fibrinogen lab test results reported during this time range. For all of the temporal trend plots, we only show data points for which there are at least 3 patients in the COVID_pos_ and COVID_neg_ cohorts. We have clarified this in the legends for the updated Figures 1, 2, and 3, and 4.

– The same question as above but then for the APTT and D-dimer data in Figure 2 vs. Figure 3 and magnesium in Figure 1 vs. Figure 3. I guess the cohorts are different between the two figures, maybe this should be stated in the figure legends.

The cohorts are different because the previous Figure 3 (now Figure 4) is restricted to patients with 3 or more lab tests during the study period, while the previous Figures 1 and 2 (now Figures 1, 2, 3) include patients with 1 or more lab tests during the study period. We have added this statement to the figure legends for clarification.

– Figure 3: we agree with the authors that the fibrinogen decline and platelet increase in COVID positive patients is re-emphasized in this manner. Also the increase in magnesium and decrease in alkaline phosphatase seem to stand out. Could the authors comment on this?

We agree that the increases in magnesium and alkaline phosphatase do seem significant. We are planning to explore this trend in future analyses.

– Could the authors comment on the number of thrombotic events that were radiographically-confirmed?

We do not have the radiology reports available in this dataset, so unfortunately we cannot comment on the number of radiographically confirmed events.

– Figure 5: for the individual patients could the authors comment on the heparin therapy with regard to the source of heparin (LMWH vs. UFH) and dosing (prophylactic vs. therapeutic)?

We have added annotations to Figure 5 indicating which medications were prescribed for prophylaxis. In addition, we have updated the Results section to include a note on these results.

– Figure 6 is very hard to understand; please find another way to graphically display the findings in a clear manner.

In order to keep the paper focused on clinical data, we have now excluded the previous Figure 6, which was based on single cell data.

Specific statistical comments:1) Cohort identification and description. (a) For the 1.3 million lab tests on 194 assays over the 60-day window: provide the mean/range of number of tests separately for positive and negative patients, and also provide the mean/range of the number of tests for pre- vs. post-index "0" date; (b) For positive patients, the proportion of those whose PCR positive test is the first test in the 60-day window; (c) If at all possible, COVID-19 related info at the first PCR test for all patients; (4) If possible, reasons for hospitalizations at the day=0;

We include Tables 7 and 8 which contain the relevant numbers for (1) (a)-(b); this should hopefully provide some further context. For PCR-positive and PCR-negative patient cohorts, we include the overall number of tests and number of patients with tests; we do this for all the tests in aggregate and 5 coagulation-related tests in particular. Table 7 has these numbers for the PCR-tested patient population; Table 8 has them for the study population of PCR-tested patients with longitudinal test data. We included general demographic characteristics as well in Table 6. We were not able to address (c) further or (d) at this time unfortunately.

2) It is not clear why it is important to assess lab tests in relation to diagnosis. For studying association between lab tests with diagnosis, ideally, the positive cases should use the "time of infection" as time zero which of course is intractable. But before variation in time from infection to diagnosis by PCR tests, the relevance of test results for diagnosis is unclear. Further, because of the propensity score matching, the cohort may not be suitable for assess the diagnosis: if some matching variables are associated with any of the test results, matching will artificially deflate the association. We therefore suggest the focus of this paper on prognosis only. The Abstract indeed focused on the prognosis, but diagnosis was mentioned multiple times in the manuscript.

We have clarified that the focus of this paper on prognosis rather than diagnosis. Furthermore, only explicit statements “at the time of diagnosis”, “after diagnosis”, or “following diagnosis” are retained in the Discussion and Results section. This should address the key concern from a statistical analysis standpoint above.

3) BERT method: The accuracy of BERT was established in an unpublished manuscript developed by the same research group. What is the proportion of patients who were classified as "Maybe", "Yes" AND "No"? It is helpful if chart review is performed on 100, say, patients, to validate this algorithm in the study context;

We have added in validation statistics for the BERT model on a randomly selected set of 1,000 sentences with thrombotic event phenotypes (see Table 5). Overall, we observe that the BERT model has an out-of-sample accuracy of 94.7%, recall of 97.8%, and precision of 92.8%. We have added a paragraph to the Materials and methods section about this validation procedure and the results.

4) Tables 3 and 4 indicate that the endpoints were significantly enriched for positive cases who had longitudinal data. But it is possible that patients received more tests because of indications of more negative outcomes. As a result, patients with longitudinal lab tests are representative of all positive patients as indicated by the enrichment. This is an important weakness of the study as this may be a source of bias, making the study results not generalizable. It is important to provide additional information on the comparing characteristics of patients with and without longitudinal tests. It may be worthwhile to provide information on reasons of repeating tests for negative patients.

We have added in a table showing the demographics of the patients who received PCR testing and our study population of patients who received PCR testing and took at least one lab test on 3+ separate days (Table 6A and B respectively). In addition, we agree with the reviewer that the study cohort of patients with longitudinal lab tests is enriched for patients with more severe disease outcomes. However, in this revision we have added in medical history clinical covariates to control for in the propensity score matching step of this analysis, so we believe that the COVID_pos_ and COVID_neg_ (matched) to be similar in their propensity for more severe disease outcomes.